# A Comprehensive Study on the Effect of Inhomogeneous Heat Dissipation on Battery Electrochemical Performance

**Yi Xie ***, **Xingyu Mu** , **Zhongwei Deng ***, **Kaiqing Zhang, Bin Chen and Yining Fan**

College of Mechanical and Vehicle Engineering, Chongqing University, Chongqing 400044, China
* Correspondence: claudexie@cqu.edu.cn (Y.X.); dengzhongw@cqu.edu.cn (Z.D.)

**Abstract:** In this paper, the unbalanced discharge of lithium-ion battery module caused by heat dissipation is studied. The battery pack is composed of 12 batteries, which are divided into four modules in series, and three batteries in each module are in parallel. The three-dimensional electrochemical-thermal model of a single battery and a battery pack is established by the polynomial approximation pseudo-two-dimensional (PP2D) method in ANSYS fluent 16.0, and the correctness of the model is verified by simulation and experiment. On this basis, the non-uniform temperature distribution and the coupling relationship between electrical parameters and electrochemical parameters in the battery pack under inhomogeneous heat dissipation were studied. The mechanism of how the temperature difference affects the distribution of current and state of charge (SOC) is also given. According to the research results, the control of the average temperature of the battery pack and the control of temperature difference are incompatible and need to be traded off. Enhanced cooling can reduce the average temperature, but it produces a large temperature gradient, resulting in a greater internal temperature difference of the battery pack. The large temperature difference enlarges the difference of the branch current and aggravates the unevenness of SOC in the battery pack. In addition, the temperature difference most suitable for SOC uniformity is not the traditional 5 °C but increases with the increase of the ambient temperature and the cooling medium temperature.

**Keywords:** lithium-ion battery module; non-uniformity; current distribution; electrochemical-thermal coupling model





## 1. Introduction

Due to the issues of global warming and the aggravation of the energy crisis, transportation electrification has become a key concern for many countries. Electric vehicles (EVs), as its key component, are becoming popular all over the world. Because lithium-ion batteries (LIBs) have high power density, large energy density, and a long lifespan, they are the main power source of EVs [1–3]. In the daily use of LIBs, their temperature greatly affects their electrical performance. The overly high temperature makes batteries degrade quickly and even evokes the thermal runaway, and the excessively low temperature greatly weakens the battery electrical performance. Thus, the battery-based energy storage system needs a thermal management system to improve its performance, maintain its security, and extend its lifespan.

Battery cells (referred to as cells) usually form a battery pack through series and a parallel connection to achieve the high power output and long driving range required by EVs [4,5]. For example, the standard range version of Tesla Model 3 has 2976 cells arranged in 96 groups of 31. This connection method often produces the inhomogeneity among cells in the pack. The inconsistencies of voltage and capacity are very common [6,7]. Moreover, the great temperature gradient in the pack caused by unevenly cooling or heating also evokes the differences of voltage and capacity [8,9]. These inhomogeneities finally reduce the cell's DoD (depth of discharge), enlarge the SOC gradient, and aggravate the battery degradation. Hence, the improvement of the consistency of cells is necessary. Eliminating

the temperature gradient of the pack is an important method to achieve this goal. Till now, many temperature-control methods have been studied. Chen, Choi et al. [10,11] optimized the air passageway to reduce the battery temperature and improve its homogeneity. Monika, Li, Sheng et al. [12–14] used cold plates with different channels to improve the battery thermal performance. Zhang and El et al. [15,16] enhanced the heat dissipation by adding heat pipes and phase-change materials to the battery. Although these methods successfully achieve a low battery temperature increase and control the battery temperature difference below 5 °C [17–19], they neither clarify the benefit of low temperature increase to the improvement of battery electrochemical performance nor explain why the temperature variation among cells should be below 5 °C. This is because these studies establish only the thermal model and do not connect the battery thermal performance with its electrochemical behavior. In the practical battery pack for EVs, there are hundreds of cells. It is difficult to keep the temperature gradient below 5 °C because the coolant for the battery pack is cooled or heated unevenly. Thus, the study of the effect of the battery thermal homogeneity on its electrochemical and electrical parameters for a better battery performance is necessary.

To study this influence, the electrochemical-thermal model is needed. Although Qin, Basu, Yang et al. [20–22] established the electrochemical-thermal models for the battery and used them to study relationship between the battery thermal dynamics and their electrochemical behaviors, these coupled models are built only at the cell level and ignore the electrical interaction among cells in the parallel-connected pack, which cannot represent the distributions of the current and the SOC caused by the uneven cooling or heating. The loss of SOC distribution further results in the prediction distortion of the electrochemical kinetics and finally brings the big prediction error to the battery performance. Moreover, the P2D (pseudo-two-dimensional) model in these studies is structurally complicated and needs a lot of computation for solving. Thus, it should be improved to reduce the computation cost and increase the robustness. In order to fill the study gap, an electrochemical-electrical-thermal model is established to investigate the inconsistencies of the electrical and electrochemical parameters. This paper has three contributions. Firstly, the two-layered model is proposed for the battery pack. The current distribution is calculated by Kirchhoff's law and coupled with the battery terminal voltage model to form the electrical model at the battery pack level. It is combined with the pack thermal model to show the influence of the inhomogeneous temperature distribution inside the pack on current difference in the parallel-connected branch and the SOC consistency among cells in the pack. At the cell level, the electrochemical model based on the PP2D model (polynomial approximation pseudo-two-dimensional model (PP2D)) is set up and coupled with the thermal model. The PP2D model employs polynomial functions to approximate the electrochemical parameters. It can not only maintain the high prediction accuracy but can also achieve much smaller computation complexity than the traditional P2D model. Moreover, with its help, the influence of the SOC distribution and the temperature gradient in the pack on the electrochemical parameters inside the cell can be known, which further reveals the interaction mechanism among the electrical, thermal, and electrochemical parameters. Secondly, the interaction among the temperature, the electrical parameter, and the electrochemical parameter inside the parallel-connected pack is quantitatively studied. Uneven heat dissipation conditions with the coolant temperature between 10 °C and 30 °C and the heat-transfer coefficient from 10 W/(K·m$^2$) to 200 W/(K·m$^2$) are imposed on the battery pack to create various temperature distributions. The distribution of the current; the SOC; and the electrochemical parameters such as electronic concentration, potential difference, local volumetric current density, and so on are computed to quantitatively explain the effect of the temperature distribution among cells on the electrical and electrochemical performance of the battery and find a suitable thermal parameter to improve the electrical and electrochemical performance. Thirdly, the ambient temperatures vary from 10 °C to 30 °C are applied. They are combined with cases of uneven cooling to achieve the suitable range of the temperature gradients in the pack at different ambient temperatures for good consistencies of SOC electrochemical parameters among cells.

## 2. Electro-Electrochemical-Thermal Model of Battery Pack

Figure 1 shows the structure of the electro-electrochemical-thermal model for the battery pack. At the cell level, the PP2D model is used to describe the electrochemical behavior inside the cell. It also provides the electrochemical parameter to compute the heat generation inside cell. At the pack level, the current distribution model is established, and it is coupled with the cell electrochemical model to form the electro-electrochemical model of the pack. This coupled model can not only predict the current distribution when the cells are series-parallel connected but also describe the effect of the uneven current distribution on the electrochemical parameters in cells. The cell thermal model is coupled with the heat transfer model at the pack level to form the pack thermal model. This model is combined with the electro-electrochemical model of pack to demonstrate the interaction among the temperature gradient inside the pack, the current distribution, and the electrochemical behavior of the cell.

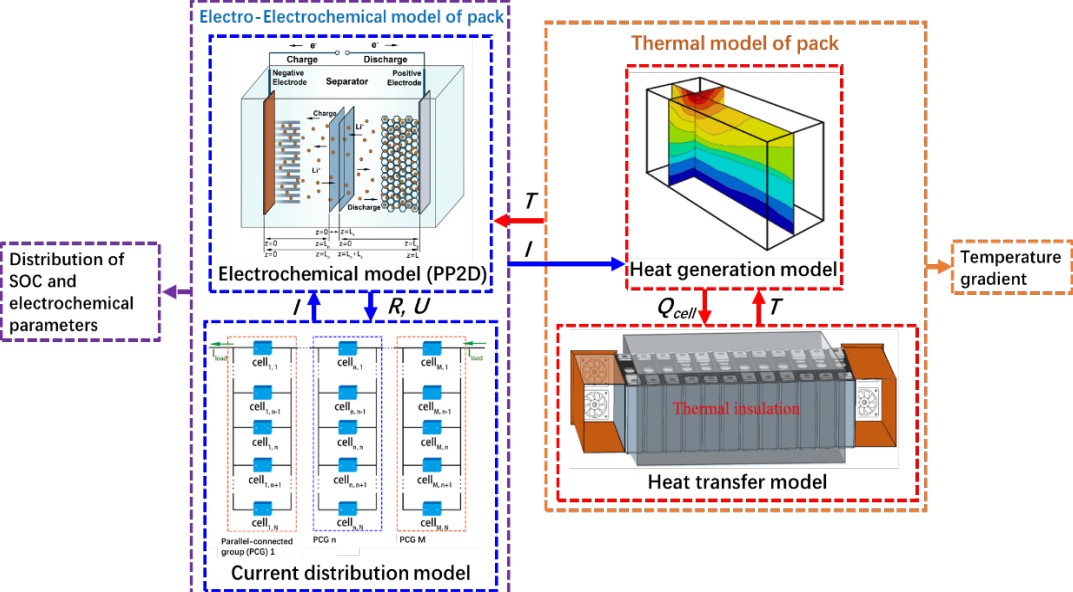

**Figure 1.** Structure of electro-electrochemical-thermal model for battery pack.

### 2.1. Cell Model

2.1.1. Electrochemical Model

The pseudo two-dimensional model (P2D) is a strictly physics-based model and describes the electrochemical process in the batteries. Table 1 shows the structure of the P2D model, and the meanings of the variables are given in the "Nomenclature". The $x_{P2D}$ coordinate is used for the P2D model at the microscale, and its direction shown in Figure 2 is from the negative electrode to the positive electrode. The additional pseudo dimension $r$ for the P2D model is along the radial distance of the electrode particle $r$.

The high-order partial differential equations (PDEs) involved in the P2D model require complex calculation, which makes the application difficult. Therefore, it is meaningful and necessary for the electrochemical model to be simplified and reduce the loss of model accuracy. The polynomial has a simple structure and can maintain high accuracy. Moreover, solving it requires little calculation. Therefore, we use the polynomial to approximate the solutions of the PDEs above, and the P2D model becomes the polynomial P2D model or PP2D model. Another major difference between the P2D model and the PP2D model lies in the selection of the coordinate axis. In order to easily calculate the PP2D model and apply it to each area of the cell, the $z_{P2D}$ coordinate is used to replace the $x_{P2D}$ coordinate in Table 1. Because $z_{P2D}$ is defined in the electrode and the diaphragm areas, respectively, axis $z_{P2D}$ is divided into $z_{P2D,n}$ for the negative electrode, $z_{P2D,s}$ for the separator, and $z_{P2D,p}$ for

the positive electrode. Figure 2 shows the regions of $z_{P2D,n}$, $z_{P2D,s}$ and $z_{P2D,p}$, and Table 2 shows the P2D model in the $z_{P2D}$ coordinate.

**Table 1.** Structure of P2D model.

| Negative/Positive Electrode | |
|---|---|
| Governing equations | Boundary conditions |

Governing equations (1):

$$\frac{\partial c_s}{\partial t} = \frac{D_s}{r^2}\frac{\partial}{\partial r}\left(r^2\frac{\partial c_s}{\partial r}\right)$$

$$\varepsilon_e \frac{\partial c_e}{\partial t} = D_e^{eff}\frac{\partial^2 c_e}{\partial x_{P2D}^2} + (1-t^+)\frac{j_f}{F}$$

$$\sigma^{eff}\frac{\partial^2 \varphi_s}{\partial x_{P2D}^2} = j_f$$

$$\kappa^{eff}\frac{\partial^2 \varphi_e}{\partial x_{P2D}^2} + \frac{2RT\kappa^{eff}(t^+-1)}{F}\left(1+\frac{d\,In\,f_\pm}{d\,In\,C_e}\right)\frac{\partial^2 Inc_e}{\partial x_{P2D}^2} + j_f = 0$$

$$j_f = a_s i_0\left[\exp\left(\frac{\alpha_a F}{RT}\eta\right) - \exp\left(-\frac{\alpha_c F}{RT}\eta\right)\right]$$

$$i_0 = Fkc_e^{\alpha_a}(c_{s,max}-c_{s,e})^{\alpha_a}c_{s,e}^{\alpha_c}$$

Boundary conditions (2):

$$D_s\frac{\partial C_s}{\partial r}\Big|_{r=0} = 0$$

$$D_s\frac{\partial C_s}{\partial r}\Big|_{r=R_s} = -\frac{j_f}{a_s F}$$

$$\frac{\partial c_e}{\partial x_{P2D}}\Big|_{x_{P2D}=0} = \frac{\partial c_e}{\partial x_{P2D}}\Big|_{x_{P2D}=L} = 0$$

$$-\sigma^{eff}\frac{\partial \varphi_s}{\partial x_{P2D}}\Big|_{x_{P2D}=0} = \sigma^{eff}\frac{\partial \varphi_s}{\partial x_{P2D}}\Big|_{x_{P2D}=L} = \frac{I}{A}$$

$$\frac{\partial \varphi_s}{\partial x_{P2D}}\Big|_{x_{P2D}=L_n} = \frac{\partial \varphi_s}{\partial x}\Big|_{x_{P2D}=L_n+L_s} = 0$$

$$\frac{\partial \varphi_e}{\partial x_{P2D}}\Big|_{x_{P2D}=0} = \frac{\partial \varphi_e}{\partial x_{P2D}}\Big|_{x_{P2D}=L} = 0$$

| Separator | |
|---|---|
| Governing equations | Boundary conditions |

Governing equations (3):

$$\varepsilon_e\frac{\partial c_e}{\partial t} = D_e^{eff}\frac{\partial^2 c_e}{\partial x_{P2D}^2}$$

$$\kappa^{eff}\frac{\partial^2 \varphi_e}{\partial x_{P2D}^2} + \frac{2RT\kappa^{eff}(t^+-1)}{F}\left(1+\frac{d\,In\,f_\pm}{d\,In\,c_e}\right)\frac{\partial^2 Inc_e}{\partial x_{P2D}^2} = \frac{I}{A}$$

Boundary conditions (4):

$$c_e|_{x_{P2D}=L_n^-} = c_e|_{x_{P2D}=L_n^+}$$

$$D_{e,n}^{eff}\frac{\partial C_e}{\partial x_{P2D}}\Big|_{x_{P2D}=L_n^-} = D_{e,s}^{eff}\frac{\partial C_e}{\partial x_{P2D}}\Big|_{x_{P2D}=L_n^+}$$

$$c_e|_{x_{P2D}=L_n+L_s^-} = c_e|_{x_{P2D}=L_n+L_s^+}$$

$$D_{e,s}^{eff}\frac{\partial C_e}{\partial x_{P2D}}\Big|_{x_{P2D}=L_n+L_s^-} = D_{e,s}^{eff}\frac{\partial C_e}{\partial x_{P2D}}\Big|_{x_{P2D}=L_n+L_s^+}$$

$$\varphi_e|_{x_{P2D}=L_n^-} = \varphi_e|_{x_{P2D}=L_n^+}$$

$$\kappa_{e,n}^{eff}\frac{\partial \varphi_e}{\partial x_{P2D}}\Big|_{x_{P2D}=L_n^-} = \kappa_{e,s}^{eff}\frac{\partial \varphi_e}{\partial x_{P2D}}\Big|_{x_{P2D}=L_n^+}$$

$$\varphi_e|_{x_{P2D}=L_n+L_s^-} = \varphi_e|_{x_{P2D}=L_n+L_s^+}$$

$$\kappa_{e,s}^{eff}\frac{\partial \varphi_e}{\partial x_{P2D}}\Big|_{x_{P2D}=L_n+L_s^-} = \kappa_{e,p}^{eff}\frac{\partial \varphi_e}{\partial x_{P2D}}\Big|_{x_{P2D}=L_n+L_s^+}$$

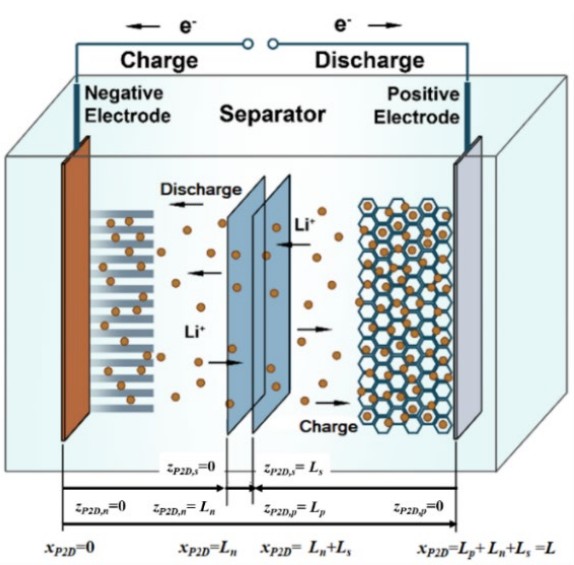

**Figure 2.** Charge transfer in the battery cell [23].

**Table 2.** Structure of P2D model in the $z_{P2D}$ coordinate.

| Negative/Positive Electrode ($k = p$ or $n$ for Positive and Negative Electrode Separately) | |
| --- | --- |
| Governing equations | Boundary conditions |
| $$\left\{ \begin{array}{c} \frac{\partial c_s}{\partial t} = \frac{D_s}{r^2}\frac{\partial}{\partial r}\left(r^2\frac{\partial c_s}{\partial r}\right) \\ \varepsilon_e\frac{\partial c_e}{\partial t} = D_e^{eff}\frac{\partial^2 c_e}{\partial z_{P2D,k}^2} + (1-t^+)\frac{j_f}{F} \\ \sigma^{eff}\frac{\partial^2 \varphi_s}{\partial z_{P2D,k}^2} = j_f \\ \kappa^{eff}\frac{\partial^2 \varphi_e}{\partial z_{P2D,k}^2} + \frac{2RT\kappa^{eff}(t^+-1)}{F}\left(1+\frac{d\,ln\,f_\pm}{d\,ln\,C_e}\right)\frac{\partial^2 lnc_e}{\partial z_{P2D,k}^2} + j_f = 0 \\ j_f = a_s i_0\left[\exp\left(\frac{\alpha_a F}{RT}\eta\right) - \exp\left(-\frac{\alpha_c F}{RT}\eta\right)\right] \\ i_0 = Fkc_e^{\alpha_a}(c_{s,max}-c_{s,e})^{\alpha_a}c_{s,e}^{\alpha_c} \end{array} \right.$$ (5) | $$\left\{ \begin{array}{c} D_s\frac{\partial C_s}{\partial r}\big|_{r=0} = 0 \\ D_s\frac{\partial C_s}{\partial r}\big|_{r=R_s} = -\frac{j_f}{a_s F} \\ \frac{\partial c_e}{\partial z_{P2D,n}}\big|_{z_{P2D,n}=0} = \frac{\partial c_e}{\partial z_{P2D,p}}\big|_{z_{P2D,p}=0} = 0 \\ -\sigma^{eff}\frac{\partial \varphi_s}{\partial z_{P2D,n}}\big|_{z_{P2D,n}=0} = \sigma^{eff}\frac{\partial \varphi_s}{\partial z_{P2D,p}}\big|_{z_{P2D,p}=0} = \frac{I}{A} \\ \frac{\partial \varphi_s}{\partial z_{P2D,n}}\big|_{z_{P2D,n}=L_n} = \frac{\partial \varphi_s}{\partial z_{P2D,p}}\big|_{z_{P2D,p}=L_p} = 0 \\ \frac{\partial \varphi_e}{\partial z_{P2D,n}}\big|_{z_{P2D,n}=0} = \frac{\partial \varphi_e}{\partial z_{P2D,p}}\big|_{z_{P2D,p}=0} = 0 \end{array} \right.$$ (6) |
| Separator | |
| Governing equations | Boundary conditions |
| $$\left\{ \begin{array}{c} \varepsilon_e\frac{\partial c_e}{\partial t} = D_e^{eff}\frac{\partial^2 c_e}{\partial z_{P2D,s}^2} \\ \kappa^{eff}\frac{\partial^2 \varphi_e}{\partial z_{P2D,s}^2} + \frac{2RT\kappa^{eff}(t^+-1)}{F}\left(1+\frac{d\,ln\,f_\pm}{d\,ln\,c_e}\right)\frac{\partial^2 lnc_e}{\partial z_{P2D,s}^2} = \frac{I}{A} \end{array} \right.$$ (7) | $$\left\{ \begin{array}{c} c_e\big|_{z_{P2D,n}=L_n} = c_e\big|_{z_{P2D,s}=0} \\ D_{e,n}^{eff}\frac{\partial C_e}{\partial z_{P2D,n}}\big|_{z_{P2D,n}=L_n} = D_{e,s}^{eff}\frac{\partial C_e}{\partial z_{P2D,s}}\big|_{z_{P2D,s}=0} \\ c_e\big|_{z_{P2D,s}=L_s} = c_e\big|_{z_{P2D,p}=L_p} \\ D_{e,s}^{eff}\frac{\partial C_e}{\partial z_{P2D,s}}\big|_{z_{P2D,s}=L_s} = -D_{e,p}^{eff}\frac{\partial C_e}{\partial z_{P2D,p}}\big|_{z_{P2D,p}=L_p} \\ \varphi_e\big|_{z_{P2D,n}=L_n} = \varphi_e\big|_{z_{P2D,s}=0} \\ \kappa_{e,n}^{eff}\frac{\partial \varphi_e}{\partial z_{P2D,n}}\big|_{z_{P2D,n}=L_n} = \kappa_{e,s}^{eff}\frac{\partial \varphi_e}{\partial z_{P2D,s}}\big|_{z_{P2D,s}=0} \\ \varphi_e\big|_{z_{P2D,s}=L_s} = \varphi_e\big|_{z_{P2D,p}=L_{P2D,p}} \\ \kappa_{e,s}^{eff}\frac{\partial \varphi_e}{\partial z_{P2D,s}}\big|_{z_{P2D,s}=L_s} = \kappa_{e,p}^{eff}\frac{\partial \varphi_e}{\partial z_{P2D,p}}\big|_{z_{z_{P2D,p}}=L_p} \end{array} \right.$$ (8) |

(a)   Electrolyte concentration approximation

Because the Li-ion diffusion equation given in Equation (5) is the second-order PDE, the distribution of the electrolyte concentration along the $z$ axis is assumed to be parabolic. The second-order polynomials are used for the concentration distributions in the electrolyte and separator, and they are

$$\left\{ \begin{array}{c} c_{e,n}(z_{P2D,n}) = a_1 z_{P2D,n}{}^2 + a_{1,1}z_{P2D,n} + a_0, 0 \leq z_{P2D,n} \leq L_n \\ c_{e,s}(z_{P2D,s}) = a_4 z_{P2D,s}{}^2 + a_3 z_{P2D,s} + a_2, 0 \leq z_{P2D,s} \leq L_s \\ c_{e,p}(z_{P2D,p}) = a_6 z_{P2D,p}{}^2 + a_{3,1}z_{P2D,p} + a_5, 0 \leq z_{P2D,p} \leq L_p \end{array} \right. . \tag{9}$$

In Equation (9), $c_e$ is the electrolyte concentration and the subscript n represents the negative electrode. The subscript $s$ represents the separator, and the subscript $p$ represents the positive electrode. In order to satisfy the boundary condition of the Li-ion diffusion equation given the third equation of Equation (6), the coefficients $a_{1,1}$ and $a_{3,1}$ are 0, and Equation (9) is simplified as

$$\left\{ \begin{array}{c} c_{e,n}(z_{P2D,n}) = a_1 z_{P2D,n}{}^2 + a_0, 0 \leq z_{P2D,n} \leq L_n \\ c_{e,s}(z_{P2D,s}) = a_4 z_{P2D,s}{}^2 + a_3 z_{P2D,s} + a_2, 0 \leq z_{P2D,s} \leq L_s \\ c_{e,p}(z_{P2D,p}) = a_6 z_{P2D,p}{}^2 + a_5, 0 \leq z_{P2D,p} \leq L_p \end{array} \right. . \tag{10}$$

The coefficients $[a_0, a_1, a_2, a_3, a_4, a_5, a_6]$ in Equation (10) are time-varying parameters, which need to be calculated in each time step. Substituting the first four equations in Equation (8) with Equation (10) gives

$$a_1 L_n^2 + a_0 = a_2 \tag{11}$$

$$2a_1 L_n D_{e,n}^{eff} = a_3 D_{e,s}^{eff} \tag{12}$$

$$a_6 L_p^2 + a_5 = a_4 L_s^2 + a_3 L_s + a_2 \tag{13}$$

$$-2a_6 L_p D_{e,p}^{eff} = (2a_4 L_s + a_3) D_{e,s}^{eff} \tag{14}$$

where $D_e$ is the liquid diffusion coefficient, $D_e{}^{eff}$ is the effective liquid diffusion coefficient, and their relationship is $D_e = D_e{}^{eff}\varepsilon_e{}^{-1.5}$. $\varepsilon_e$ is the volume fraction of Li-ion in the electrolyte, and $\varepsilon_e c_e$ is the concentration of Li-ion. The mole number of Li-ion at the unit area in the regions of the negative electrode $Q_{e,n}(t)$, the separator $Q_s(t)$, and the positive electrode $Q_{e,p}(t)$ are obtained by integrating its concentration along the $z$-axis, and they are

$$Q_{e,n}(t) = \int_0^{L_n} \varepsilon_{e,n} c_{e,n}(z_{P2D,n}) dz_{P2D,n} = \varepsilon_{e,n}\left(\frac{1}{3}a_1 L_n^3 + a_0 L_n\right) \tag{15}$$

$$Q_{e,s}(t) = \int_0^{L_s} \varepsilon_{e,s} c_{e,s}(z_{P2D,s}) dz_{P2D,s} = \varepsilon_{e,s}\left(\frac{1}{3}a_4 L_s^3 + \frac{1}{2}a_3 L_s^2 + a_2 L_s\right) \tag{16}$$

$$Q_{e,p}(t) = \int_0^{L_s} \varepsilon_{e,p} c_{e,p}(z_{P2D,p}) dz_{P2D,p} = \varepsilon_{e,p}\left(\frac{1}{3}a_6 L_p^3 + a_5 L_p\right). \tag{17}$$

The surface density of the current at the electrode ($I/A$) can be achieved by integrating the volumetric current density $j_f$ along the $z$ axis, and the expressions are

$$\int_0^{L_n} j_{f,n}(z_{P2D,n}) dz_{P2D,n} = \frac{I}{A} \tag{18}$$

$$\int_0^{L_p} j_{f,p}(z_{P2D,p}) dz_{P2D,p} = -\frac{I}{A} \tag{19}$$

where $A$ means the electrode plate area and $I$ is the current through the cell.

Substituting the second equation in Equation (5) and the first equation in Equation (7) with Equation (10), integrating these equations along the $z$ axis, and combing them with Equations (15)–(19) gives the expressions of $Q_{e,n}(t)$, $Q_{e,s}(t)$, and $Q_{e,p}(t)$, which are

$$\frac{d}{dt} Q_{e,n}(t) = (1 - t^+)\frac{I}{AF} + 2a_1 L_n D_{e,n}^{eff} \tag{20}$$

$$\frac{d}{dt} Q_{e,s}(t) = 2a_4 L_s D_{e,s}^{eff} \tag{21}$$

$$\frac{d}{dt} Q_{e,p}(t) = -(1 - t^+)\frac{I}{AF} + 2a_6 L_p D_{e,p}^{eff}. \tag{22}$$

In Equations (20)–(22), $t$ is the time and $t^+$ is the transfer number of Li-ion.

Because the initial electrolyte concentration is evenly distributed in regions of the electrodes and separator, the initial total mole numbers of Li-ion $Q_{e,n}(0)$, $Q_{e,s}(0)$, and $Q_{e,p}(0)$ can be achieved by solving

$$\begin{cases} Q_{e,n}(0) = \varepsilon_{e,n} c_{e,0} L_n \\ Q_{e,s}(0) = \varepsilon_{e,s} c_{e,0} L_s \\ Q_{e,p}(0) = \varepsilon_{e,p} c_{e,0} L_p \end{cases} \tag{23}$$

where $c_{e,0}$ is the initial electrolyte concentration. In order to satisfy Equation (23), the initial polynomial coefficients $[a_0, a_1, a_2, a_3, a_4, a_5, a_6]$ in Equations (15)–(17) should be meet the requirement below.

$$[a_0, a_1, a_2, a_3, a_4, a_5, a_6] = [c_{e,0}, 0, c_{e,0}, 0, 0, c_{e,0}, 0]. \tag{24}$$

The forward Euler method is used to solve the Equations (20)–(22) to obtain the $Q_{e,n}(t)$, $Q_{e,s}(t)$, and $Q_{e,p}(t)$ in each time step. Then, these total mole numbers of Li-ion are used for Equations (11)–(17) to calculate $[a_0, a_1, a_2, a_3, a_4, a_5, a_6]$. After the coefficient matrix is achieved, we can employ Equation (10) to approximate the distribution of the electrolyte concentration in the electrodes and separator.

(b)    Reaction flux approximation

The reaction flux $j$ is non-uniform, especially when the large current rate is loaded on the cell. The local volumetric current density $j_f$ is used to describe its distribution between the negative and positive electrodes. Because $j_f$ is distributed like a parabola, a quadratic polynomial is applied to approximating the $j_f$. It is

$$\begin{cases} j_f(z_{P2D,n}) = c_{2,n}z_{P2D,n}^2 + c_{1,n}z_{P2D,n} + c_{0,n} \ 0 \leq z_{P2D,n} \leq L_n, \text{ in negative electrode} \\ j_f(z_{P2D,p}) = c_{2,p}z_{P2D,p}^2 + c_{1,p}z_{P2D,p} + c_{0,p} \ 0 \leq z_{P2D,n} \leq L_s, \text{ in positive electrode} \end{cases}. \quad (25)$$

In the electrode, the activation overpotential $\eta$ is calculated by the potential balance equation, which is

$$\eta(z_{P2D,k}) = \varphi_s(z_{P2D,k}) - \varphi_e(z_{P2D,k}) - U(z_{P2D,k}) - \frac{R_{SEI}}{a_s}j_f(z_{P2D,k}). \quad (26)$$

In Equation (26), $\varphi_s$ is the potential of the solid phase. $U$ is the electrode equilibrium potential, and $R_{SEI}$ represents the resistance of SEI film. $\varphi_e$ represents the potential of the electrolyte. $a_s$ means the specific interfacial surface area, and the subscript $k$ is $n$ for the negative electrode or $p$ for the positive electrode. $\eta$ can also be calculated by solving the Bulter–Volmer given in Equation (5), and it is

$$\eta(z_{P2D,k}) = \frac{2RT}{F}\text{arcsinh}\left[\frac{j_f(z_{P2D,k})}{2a_s i_0}\right] \quad (27)$$

where $T$ is cell temperature, $R$ is the gas constant, $F$ is Faraday constant, and $i_0$ represents the exchange current density. Substituting the $\eta$ in Equation (26) with Equation (27) gives

$$P(z_{P2D,k})\frac{\partial j_f(z_{P2D,k})}{\partial z_{P2D,k}} = \frac{\partial \varphi_s(z_{P2D,k})}{\partial z_{P2D,k}} - \frac{\partial \varphi_e(z_{P2D,k})}{\partial z_{P2D,k}} - \frac{\partial U(z_{P2D,k})}{\partial z_{P2D,k}} \quad (28)$$

where $P(Z_{P2D,k})$ is

$$P(z_{P2D,k}) = \frac{RT}{a_s F^2 i_0}\frac{1}{\sqrt{1 + \left[\frac{j_f(z_{P2D,k})}{2a_s F i_0}\right]^2}} + \frac{R_{SEI}}{a_s}. \quad (29)$$

Integrating the third and fourth equations in Equation (5) along the $Z_{P2D,n}$ and $Z_{P2D,p}$, respectively, gives the expressions of the $\varphi_s$ and $\varphi_e$. They are

$$\frac{\partial \varphi_s(z_{P2D,k})}{\partial z_{P2D,k}} = \frac{1}{\sigma^{eff}}\int_0^{z_{P2D,k}} j_f dz_{P2D,k} - \frac{I}{\sigma^{eff} A} \quad (30)$$

$$\frac{\partial \varphi_e(z_{P2D,k})}{\partial z_{P2D,k}} = -\frac{2RT(t^+ - 1)}{F}\frac{\partial \ln c_{e,k}(z_{P2D,k})}{\partial z_{P2D,k}} - \frac{1}{\kappa^{eff}}\int_0^{z_{P2D,k}} j_f dz_{P2D,k}. \quad (31)$$

where $\sigma^{eff}$ means solid-phase effective conductivity, $t^+$ is the transfer number of lithium-ion, and $\kappa^{eff}$ is the effective conductivity of the electrolyte.

With the Equations (25)–(31) and their corresponding boundary conditions, the coefficients in Equation (25) can be solved. We take the negative electrode as an example to show how $c_{2,n}$, $c_{1,n}$ and $c_{0,n}$ in Equation (25) are reached. The subscript $k$ in Equations (26)–(31) becomes $n$. Substituting $[\partial \varphi_s(Z_{P2D,n})/\partial Z_{P2D,n}]$ and $[\partial \varphi_e(Z_{P2D,n})/\partial Z_{P2D,n}]$ in Equation (28) with Equations (30) and (31) gives

$$P(z_{P2D,n})\frac{\partial j_f(z_{P2D,n})}{\partial z_{P2D,n}} = \left[\frac{1}{\sigma^{eff}}\int_0^{z_{P2D,n}} j_f dz_{P2D,n} - \frac{I}{\sigma^{eff} A}\right] - \left[\frac{2RT(t^+ - 1)}{F}\frac{\partial \ln c_{e,n}(z_{P2D,n})}{\partial z_{P2D,n}} - \frac{1}{\kappa^{eff}}\int_0^{z_{P2D,n}} j_f dz_{P2D,n}\right] - \frac{\partial U(z_{P2D,n})}{\partial z_{P2D,n}} \quad (32)$$

where $P(Z_{P2D,n})$ is

$$P(z_{P2D,n}) = \frac{RT}{a_s F^2 i_0} \frac{1}{\sqrt{1 + \left[\frac{j_f(z_{P2D,n})}{2a_s F i_0}\right]^2}} + \frac{R_{SEI}}{a_s}. \tag{33}$$

When $Z_{P2D,n}$ is 0, Equation (32) is solved by replacing $j_f(Z_{P2D,n})$ with Equation (25), and the result is

$$P(0)c_{1,n} = -\frac{I}{\sigma^{eff} A} - \frac{\partial U(z_{P2D,n})}{\partial z_{P2D,n}}\Big|_{z_{P2D,n}=0}. \tag{34}$$

When $Z_{P2D,n}$ is $L_n$, the difference between the first second term on right side of Equation (32) is 0 according to Equation (28) and the fifth equation in Equation (6), and the integral value in the four term is $(I/A)$ according to Equation (18). With the calculation above, Equation (32) under the condition of $Z_{P2D,n} = L_n$ becomes

$$P(L_n)(2c_{2,n}L_n + c_{1,n}) = -\frac{2RT(t^+ - 1)}{F}\frac{\partial \ln c_e(z_{P2D,n})}{\partial z_{P2D,n}}\Big|_{z_{P2D,n}=L_n} + \frac{I}{\kappa^{eff} A} - \frac{\partial U(z_{P2D,n})}{\partial z_{P2D,n}}\Big|_{z_{P2D,n}=L_n}. \tag{35}$$

In order to obtain $c_{2,n}$ and $c_{1,n}$ in Equations (34) and (35), it is necessary to calculate the electrode equilibrium potential $U(Z_{P2D,n})$ at $Z_{P2D,n} = 0$ and $Z_{P2D,n} = L_n$. According to Ref. [24], $U(Z_{P2D,n})$ in the negative electrode is decided by the stoichiometry of the electrode concentration $\theta_n$, and it is

$$U(z_{P2D,n}) = 0.6554 - 5.8181\theta_n(z_{P2D,n}) + 22.5962\theta_n(z_{P2D,n})^2 - 36.1670\theta_n(z_{P2D,n})^3 + 20.0406\theta_n(z_{P2D,n})^4 \tag{36}$$

where $\theta_n(Z_{P2D,n})$ is

$$\theta_n(z_{P2D,n}) = \frac{c_{s,e}(z_{P2D,n})}{c_{s,\max}}. \tag{37}$$

The third equation for $c_{2,n}$, $c_{1,n}$, and $c_{0,n}$ in Equation (25) is achieved by integrating Equation (18), and it is

$$\frac{c_{2,n}}{3}L_n^3 + \frac{c_{1,n}}{3}L_n^2 + c_{0,n}L_n = \frac{I}{A}. \tag{38}$$

By solving Equations (34), (35), and (38), we have the coefficients of $c_{2,n}$, $c_{1,n,}$ and $c_{0,n}$ in Equation (25) and the local volumetric current density in the negative electrode. Because the cell is in a stable state in the beginning of discharging or charging, the $j_f$ is evenly distributed and the initial $j_f$ for Equation (25) is

$$j_f(z_{P2D,n}) = \frac{I}{AL}. \tag{39}$$

The calculation process of $j_f$ in the positive electrode is almost same as that in the negative electrode. The only difference is that the current $I$ in the positive electrode should be multiplied by $-1$ because the direction of the $Z_{P2D,p}$-axis is opposite to that of the $Z_{P2D,n}$-axis.

(c)　Solid-phase surface concentration approximation

According to Ref. [25], the solid-phase surface concentration of lithium-ion $c_{s,e}(t)$ is not only related to its volume-averaged concentration $\bar{c}_s(t)$ but also to its volume-averaged concentration flux $\bar{q}(t)$, and it can be calculated through [25]

$$35\frac{D_s}{R_s}[c_{s,e}(t) - \bar{c}_s(t)] - 8D_s\bar{q}(t) + \frac{j_f(z_{P2D,i})}{a_s F} = 0 \tag{40}$$

where $\bar{c}_s(t)$ and $\bar{q}(t)$ are

$$\frac{d}{dt}\bar{c}_s(t) + 3\frac{j_f(z_{P2D,i})}{a_s F R_s} = 0 \tag{41}$$

$$\frac{d}{dt}\overline{q}(t) + 30\frac{D_s}{R_s^2}\overline{q}(t) + \frac{45}{2}\frac{j_f(z_{P2D,i})}{a_s F R_s^2} = 0. \tag{42}$$

In Equations (40)–(42), $t$ is time, $D_s$ is the solid-phase diffusivity of lithium-ion, $R_s$ is the particle radius, and the subscript $i$ is $n$ for the negative electrode or $p$ for the positive electrode.

When $t = 0$, the cell is about to charge or discharge, and the initial value of $\overline{q}(0)$ is 0. Moreover, the initial value of $\overline{c}_s(0)$ can be achieved by solving

$$\overline{c}_{s,n}(0) = [\theta_0^n + SOC_{ini}(\theta_{100}^n - \theta_0^n)]c_{s,\max}^n \tag{43}$$

$$\overline{c}_{s,p}(0) = \left[\theta_0^p + SOC_{ini}\left(\theta_{100}^p - \theta_0^p\right)\right]c_{s,\max}^p \tag{44}$$

where $\overline{c}_{s,n}(0)$ and $\overline{c}_{s,p}(0)$ are the volume-averaged concentration of lithium-ion in the positive and negative electrodes; $SOC_{ini}$ is the initial SOC, $c^n_{s,max}$ and $c^p_{s,max}$ are the maximum solid-phase concentration of lithium-ion in the positive and negative electrodes; $\theta$ is the stoichiometry of electrode concentration; superscripts $n$ and $p$ represent the positive and negative electrode, and subscript 0 and 100 are 0% and 100% of the cell SOC, respectively.

(d)　Other parameters for PP2D model

This section shows how the cell terminal voltage is calculated. Moreover, the calculation methods of the variables and the constants for the PP2D model are also given in this section. Integrating Equation (31) in the electrodes and separator and using the third equation in Equation (6) as the boundary condition for Equation (31) gives the formula of the electrolyte potential, which is

$$\begin{cases} \varphi_{e,n}(z_{P2D,n}) - \varphi_{e,n}(0) = \frac{2RT(1-t^+)}{F}ln\frac{c_{e,n}(z_{P2D,n})}{c_{e,n}(0)} - \frac{1}{\kappa_n^{eff}}\int_0^{Z_{P2D,n}}\int_0^{Z_{P2D,n}} j_f(\xi)d\xi dz_{P2D,n}, & \text{in the negative electrode} \\ \varphi_{e,p}(z_{P2D,p}) - \varphi_{e,p}(0) = \frac{2RT(1-t^+)}{F}\ln\frac{c_{e,p}(z_{P2D,p})}{c_{e,p}(0)} - \frac{1}{\kappa_p^{eff}}\int_0^{Z_{P2D,p}}\int_0^{Z_{P2D,p}} j_f(\xi)d\xi dz_{P2D,p}, & \text{in the positive electrode} \\ \varphi_{e,s}(z_{P2D,s}) - \varphi_{e,s}(0) = \frac{2RT(1-t^+)}{F}\ln\frac{c_e(z_{P2D,s})}{c_e(0)} - \frac{Iz_{P2D,s}}{A\kappa_s^{eff}}, & \text{in the separator} \end{cases} \tag{45}$$

where

$$\int_0^{z_{P2D,n}}\int_0^{z_{P2D,n}} j_f(\xi)d\xi dz_{P2D,n} = \frac{c_{2,n}}{12}z_n^4 + \frac{c_{1,n}}{6}z_n^3 + \frac{c_{0,n}}{2}z_n^2 \tag{46}$$

$$\int_0^{z_{P2D,p}}\int_0^{z_{P2D,p}} j_f(\xi)d\xi dz_{P2D,p} = \frac{c_{2,p}}{12}z_p^4 + \frac{c_{1,p}}{6}z_p^3 + \frac{c_{0,p}}{2}z_p^2. \tag{47}$$

The potential difference can be achieved through Equation (45). It is

$$\varphi_{e,p}(0) - \varphi_{e,n}(0) = \frac{2RT(1-t^+)}{F}\ln\frac{c_{e,p}(0)}{c_{e,n}(0)} - \frac{1}{\kappa_n^{eff}}\int_0^{L_n}\int_0^{z_{P2D,n}} j_f(\xi)d\xi dz_{P2D,n} + \frac{1}{\kappa_p^{eff}}\int_0^{L_p}\int_0^{z_{P2D,p}} j_f(\xi)d\xi dz_{P2D,p} - \frac{IL_s}{A\kappa_s^{eff}}. \tag{48}$$

where $\varphi_e$ is the electrolyte potential; (0) is the current collector; and the subscript $p$ and $n$ represent positive and negative electrode separately. Terminal voltage $V$ of cell is reached by solving

$$V = \varphi_{s,p}(0) - \varphi_{s,n}(0). \tag{49}$$

where $\varphi_e$ is the solid-phase potential. Combining Equations (26), (37), (48), and (49) gives the expression of the cell terminal voltage. It is

$$V(t) = U_p(0) - U_n(0) + \eta_p(0) - \eta_n(0) + \varphi_{e,p}(0) - \varphi_{e,n}(0) + \frac{R_{SEI,p}}{a_{s,\,p}}j_{f,p}(0) - \frac{R_{SEI,n}}{a_{s,n}}j_{f,n}(0). \tag{50}$$

where "$_p(0)$" and "$_n(0)$" refer to the variable values at $Z_{P2D,p} = 0$ and $Z_{P2D,n} = 0$, respectively. A 50Ah NCM/graphite prismatic cell is used in this paper, and its electrochemical parameters are given in Tables 3 and 4.

**Table 3.** Parameter for the PP2D model of the 50Ah battery cell.

| Parameter | Negative Electrode | Separator | Positive Electrode | Reference |
|---|---|---|---|---|
| $L$ (m) | $73 \times 10^{-6}$ | $13 \times 10^{-6}$ | $61 \times 10^{-6}$ | [26] |
| $A$ (m$^2$) | / | 2.14 | / | Measured |
| $R_s$ (m) | $9.93 \times 10^{-6}$ | / | $6.32 \times 10^{-6}$ | [27] |
| $\varepsilon_s$ | 0.65 | / | 0.547 | [28,29] |
| $\varepsilon_e$ | 0.315 | 0.5307 | 0.332 | [28–30] |
| $c_{s,max}$ (mol m$^{-3}$) | 31,389 | / | 48,396 | [24] |
| $\theta_0$ | 0.01 | / | 0.955 | [24] |
| $\theta_{100}$ | 0.785 | / | 0.415 | [24] |
| $c_{e,0}$ (mol m$^{-3}$) | / | 1200 | / | [30] |
| $\alpha_a$, $\alpha_c$ | 0.5, 0.5 | / | 0.5, 0.5 | [24] |
| $t^+$ | / | 0.363 | / | [30] |
| $\sigma$ (Sm$^{-1}$) | 100 | / | 100 | [23] |
| $R_{SEI}$ ($\Omega$m) | 0 | / | 0 | [23] |
| d$lnf_{\pm}$/d$ln\, c_e$ | / | 0 | / | [30] |

**Table 4.** Variables for PP2D model of the 50Ah battery cell.

| Parameter | Equation | Reference |
|---|---|---|
| Diffusion coefficient of Li-ion in the electrode | $D_{s,p} = 10^{-14} \exp\left[-\frac{30000}{8.314}\left(\frac{1}{T} - \frac{1}{298.15}\right)\right]$ <br> $D_{s,n} = 1.4523 \times 10^{-13} \exp\left[-\frac{30000}{8.314}\left(\frac{1}{T} - \frac{1}{298.15}\right)\right]$ | [31] |
| Diffusion coefficient of Li-ions in the electrolyte | $D_e = 10^{\left(-8.43 - \frac{54}{T-229-0.005c_e} - 0.22 \times 0.001c_e\right)}$ | [31] |
| Reaction rate of the electrode | $k_n = 2 \times 10^{-11} \exp\left[-\frac{30000}{8.314}\left(\frac{1}{T} - \frac{1}{298.15}\right)\right]$ <br> $k_p = 2 \times 10^{-11} \exp\left[-\frac{30000}{8.314}\left(\frac{1}{T} - \frac{1}{298.15}\right)\right]$ | [30] |
| Ionic conductivity | $\kappa = 1.254c_e \times 10^{-4}(-8.248 + 0.05324T - 2.987 \times 10^{-5}T^2 + 0.2623 \times 10^{-3}c_e - 0.009306 \times 10^{-3}c_e T + 0.000008069 \times 10^{-3}c_e T^2 + 0.22 \times 10^{-6}c_e^2 - 0.0001765 \times 10^{-6}c_e^2 T)$ | [31] |
| State of charge (SOC) | $\theta_i = \frac{c_{e,i}}{c_{e,i,\max}}, \ (i = p, n)$ | [31] |
| Open circuit potential | $U_p = 4.3655 + 5.3596\theta_p - 23.8949\theta_p^2 + 30.4942\theta_p^3 - 12.7557\theta_p^4$ <br> $U_n = 0.6554 - 5.8181\theta_n + 22.5962\theta_n^2 - 36.1670\theta_n^3 + 20.0406\theta_n^4$ | [24] |
| Temperature derivative of open circuit potential | $\frac{dU_p}{dT} = -7.225 \times 10^{-5}$ <br> $\frac{dU_n}{dT} = 0.00305 - 0.002762\theta_n + 0.005726\theta_n^2 - 0.004453\theta_n^3$ | [24] |

### 2.1.2. Thermal Model

The cell thermal model includes not only the heating model but also the heat transfer model. By combining the Fourier's basic law of thermal conductivity with the law of energy conservation, the temperature distribution is obtained by solving

$$\rho C_p \frac{\partial T}{\partial t} = \frac{\partial}{\partial x}\left(k_x \frac{\partial T}{\partial x}\right) + \frac{\partial}{\partial y}\left(k_y \frac{\partial T}{\partial y}\right) + \frac{\partial}{\partial z}\left(k_z \frac{\partial T}{\partial z}\right) + q_v \tag{51}$$

where $\rho$ and $C_p$ are cell's density and specific heat capacity, and $T$ represents temperature. $t$ means time, and $q_v$ is the volume-averaged heat generation rate. $k_x$, $k_y$, and $k_z$ represent the thermal conductivities along the direction of $x$, $y$, and $z$ axes separately. The heat generation rate inside the cell is composed of the reaction heat generation $q_{rea}$, ohmic heat generation $q_{ohm}$, and active heat generation $q_{act}$, which are calculated by

$$q_{rea} = j_f(Z_{P2D,n})T\frac{dU(Z_{P2D,n})}{dT} + j_f(Z_{P2D,p})T\frac{dU(Z_{P2D,p})}{dT} \tag{52}$$

$$q_{act} = \frac{\int_0^{L_n} j_f(Z_{P2D,n})*[\phi_{s,n}(Z_{P2D,n})-\phi_{e,n}(Z_{P2D,n})-U(Z_{P2D,n})]dz_{P2D,n}+\int_0^{L_p} j_f(Z_{P2D,p})*[\phi_{s,p}(Z_{P2D,p})-\phi_{e,p}(Z_{P2D,p})-U(Z_{P2D,p})]dz_{P2D,p}}{L} \quad (53)$$

$$q_{ohm} = \frac{1}{L}\int_0^{L_n}\left[\sigma^{eff}\left(\frac{\partial\phi_{s,n}(Z_{P2D,n})}{\partial z_{P2D,n}}\right)^2 + \kappa_n^{eff}\left(\frac{\partial\phi_{e,n}(Z_{P2D,n})}{\partial z_{P2D,n}}\right)^2 + \kappa_D^{eff}\left(\frac{\partial\ln c_{e,n}(Z_{P2D,n})}{\partial z_{P2D,n}}\right)\left(\frac{\partial\phi_{e,n}(Z_{P2D,n})}{\partial z_{P2D,n}}\right)\right]dz_{P2D,n}+$$

$$\frac{1}{L}\int_0^{L_p}\left[\sigma^{eff}\left(\frac{\partial\phi_{s,p}(Z_{P2D,p})}{\partial z_{P2D,p}}\right)^2 + \kappa_p^{eff}\left(\frac{\partial\phi_{e,p}(Z_{P2D,p})}{\partial z_{P2D,p}}\right)^2 + \kappa_p^{eff}\left(\frac{\partial\ln c_{e,p}(Z_{P2D,p})}{\partial z_{P2D,p}}\right)\left(\frac{\partial\phi_{e,p}(Z_{P2D,p})}{\partial z_{P2D,p}}\right)\right]dz_{P2D,p}+ \quad (54)$$

$$\frac{1}{L}\int_0^{L_s}\left[\kappa_s^{eff}\left(\frac{\partial\phi_{e,s}(Z_{P2D,s})}{\partial z_{P2D,s}}\right)^2 + \kappa_s^{eff}\left(\frac{\partial\ln c_{e,s}(Z_{P2D,s})}{\partial z_{P2D,s}}\right)\left(\frac{\partial\phi_{e,s}(Z_{P2D,s})}{\partial z_{P2D,s}}\right)\right]dz_{P2D,s}.$$

Total heat generation rate in a unit volume of the cell $q_v$ is

$$q_v = \frac{q_{rea} + q_{act} + q_{ohm}}{v_b} \quad (55)$$

where $v_b$ is the cell volume.

### 2.1.3. Specific Heat Capacity and Thermal Conductivities

Experiments were carried out to measure 50 Ah cell's specific heat capacity and thermal conductivities. Figure 3a gives the experiment devices for the test of the specific heat capacity. Before $C_p$ was tested, the cell was covered with aluminum foil to enhance the heat transfer and ensure that its temperature was homogeneously distributed. The PTC heating plate was fixed on the surface of the aluminum to heat the cell. The whole device was put in the accelerating rate calorimeter (ARC), which was used to make the ambient temperature the same as the cell temperature and avoid heat transfer among cell and air. Figure 3b presents the measured cell temperature increase. The slope of the temperature line is used to calculate the $C_p$, which is expressed by

$$C_p = \frac{P_{heat}}{m_b}\frac{1}{\Delta T/\Delta t}. \quad (56)$$

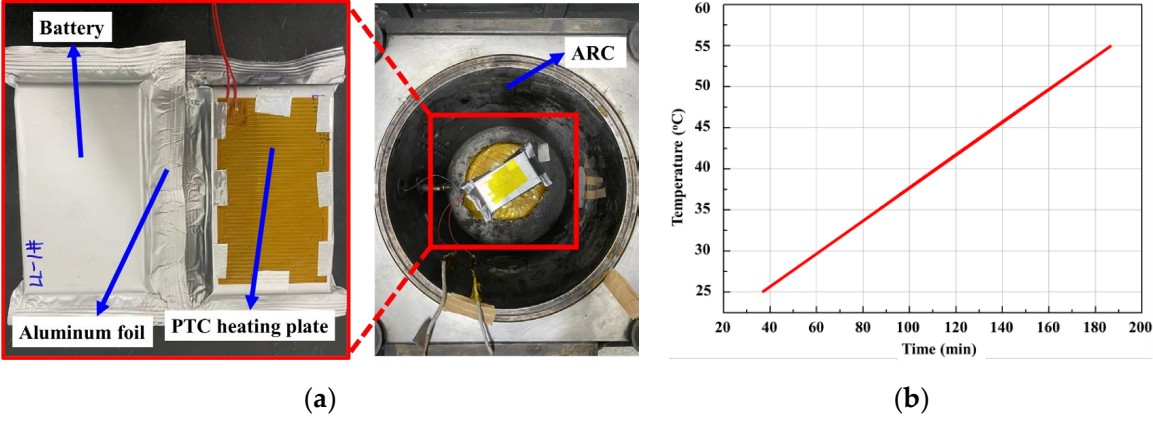

**Figure 3.** Test device for cell $C_p$ and test results: (**a**) experiment devices, (**b**) cell temperature increase.

In Equation (56), $P_{heat}$ is the heating power of the PTC, $m_b$ is the cell mass, and $\Delta T/\Delta t$ is the slope of the temperature line in Figure 3b. The tested $C_p$ of the 50 Ah prismatic cell is 989 J/(kg·K).

The thermal conductivities of the cell were measured based on the transient plane source (TPS) method [32]. Figure 4 gives the test devices for the cell thermal conductivity. The cell was covered with the aluminum foil and put in a thermostat. Ambient temperature was set as 25 °C. The sensor was fixed on the aluminum foil. It was used to heat the cell

and measure the temperature increase. The thermal conductivity test instrument (type: Hot Disk TPS 3500) provided the sensor with the heating power and measured the thermal conductivity of cell. The computer was used to display the test results. The theory about the TPS and the experiment process is presented in Ref. [33]. The measured thermal conductivity along the cell thickness $k_x$ is 1.26 W/(m·K), and the thermal conductivities along the cell width and height $k_y$ and $k_z$ are 23.36 W/(m·K).

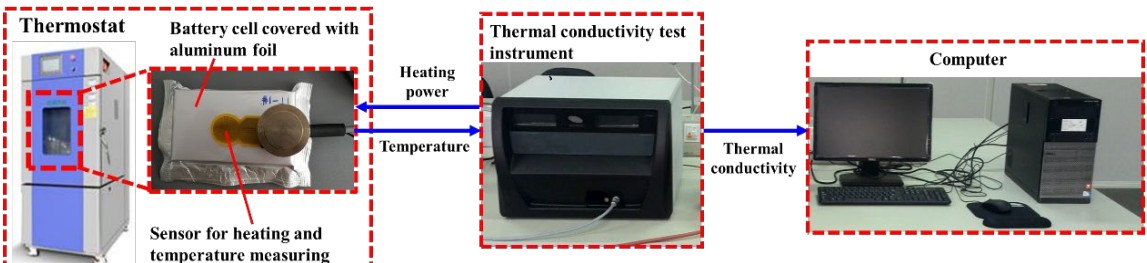

**Figure 4.** Test devices for cell thermal conductivity.

*2.2. Pack Model*

2.2.1. Current Distribution Model

Because the cell temperature has a great influence on resistance [34], the inhomogeneous temperature distribution evokes an inconsistent resistance distribution, which results in the variation in the current through each parallel-connected battery string (PCBS) and aggravates the inhomogeneous temperature distribution. To accurately calculate the temperature distribution in the pack and describe the influence of the thermal consistency among cells on the cell electrochemical performance, the current distribution at the pack level in the battery model needs to be considered. Figure 5 shows the equivalent circuit of the 3P4S pack. The total terminal voltage of the parallel battery string in PCBS_$j$ ($j$ = 1, 2, 3, and 4) is calculated by Kirchhoff's voltage law

$$V_j = V_{j,1} - rI_{j,1} = V_{j,2} - rI_{j,2} = V_{j,3} - rI_{j,3} \tag{57}$$

where $V_j$ is total terminal voltage of the PCBS_$j$, $V_{4,i}$ ($i$ = 1, 2 and 3) is the cell's terminal voltage in the PCBS_4, $I_{4,i}$ ($i$ = 1, 2 and 3) is the branch current through the cell $i$ in the PCBS_$j$, and $r$ is the welding resistance. The terminal voltage of cell $i$ in the PCBS_$j$ $V_{j,i}$ is

$$V_{j,i} = U_{OCV,j,i} - R_{j,i}I_{j,i} \tag{58}$$

where $U_{OCV}$ is the open-circuit voltage (OCV), $R$ is the resistance, and the subscript ($j,i$) is the $i$th cell in PCBS_$j$. The current distribution in the PCBS_$j$ is achieved by solving Equations (57) and (58). It is

$$\begin{bmatrix} I_{j,1} \\ I_{j,2} \\ I_{j,3} \end{bmatrix} = \begin{bmatrix} 1 & 0 & 0 \\ 0 & -R_{j,1} - R_{j,2} + 2r & R_{j,2} - r \\ 0 & R_{j,2} - r & -R_{j,2} - R_{j,3} + 2r \end{bmatrix}^{-1} \left( \begin{bmatrix} 0 & 0 & 0 \\ 1 & -1 & 0 \\ 0 & 1 & -1 \end{bmatrix} \cdot \begin{bmatrix} U_{OCV,j,1} \\ U_{OCV,j,2} \\ U_{OCV,j,3} \end{bmatrix} + \begin{bmatrix} 1 \\ r - R_{j,1} \\ 0 \end{bmatrix} I \right) \tag{59}$$

where the subscript $j$ is the $j$th PCBS or PCBS_$j$. According to the PP2D model, $U_{OCV}$ can be calculated by

$$U_{OCV} = U_p(0) - U_n(0). \tag{60}$$

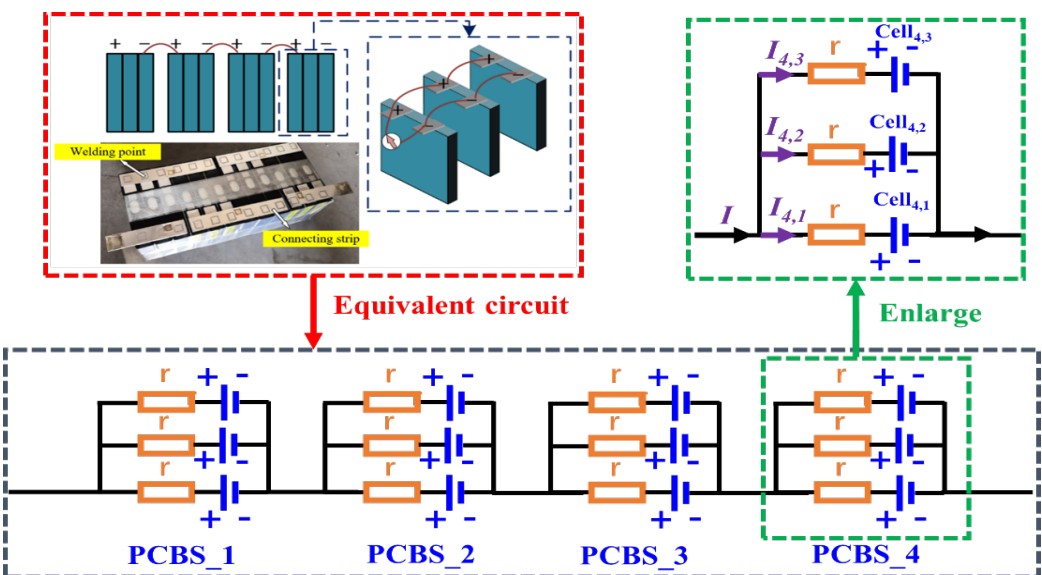

**Figure 5.** Equivalent circuit of the 3P4S pack.

In the solution process for the current distribution, the cell resistance in time step $t$ is calculated according to Equations (58)–(60). Then, they are used to calculate the current distribution in the next time step ($t + 1$).

The welding resistance $r$ is computed by pack resistance $R_{bp}$ and cell resistance $R_{j,i}$. Firstly, the $R_{bp}$ and $R_{j,i}$ are measured by the test of HPPC (hybrid pulse power characterization) at 1 C discharging rate, SOC of 100%, and 20 °C ambient temperature. Then, Equation (61) is applied to calculate the r of the 3P4S pack, and $r$ is 0.717 mΩ.

$$R_{bp} = \sum_{j=1}^{4} \frac{1}{\sum_{i=1}^{3} \frac{1}{(R_{j,i} + r)}}. \tag{61}$$

### 2.2.2. Thermal Model of Pack

Pack thermal model integrates the cell thermal model, heat generated by the welding point, and the heat transfer process of the pack. It is

$$\rho c_p \frac{\partial T}{\partial t} = \frac{\partial}{\partial x}\left(k_x \frac{\partial T}{\partial x}\right) + \frac{\partial}{\partial y}\left(k_y \frac{\partial T}{\partial y}\right) + \frac{\partial}{\partial z}\left(k_z \frac{\partial T}{\partial z}\right) + (q_v + q_w). \tag{62}$$

$q_w$ is heat production of welding point. It is

$$q_w = \frac{I_j^2 r}{V_W} \tag{63}$$

where $V_W$ represents the welding point's volume. The third kind of heat transfer condition [35] is employed for the pack thermal model, and it is

$$-k\left(\frac{\partial T}{\partial n}\right)_{pack\_wall} = h(T_w - T_c) \tag{64}$$

where $k$ represents thermal conductivity of the coolant. $\frac{\partial T}{\partial n}$ indicates the vertical temperature gradient close to the pack surface, and $T_w$ means the temperature of the pack surface. $T_c$ is the coolant temperature close to the pack. $h$ represents the heat-transfer coefficient.

### 2.3. Solution of Electrochemical-Thermal Model for the Pack

The FEM (finite element method) is applied to obtain the influence of the inhomogeneous temperature distribution on the electrochemical performance. Figure 6 gives the solution process of the model. Firstly, the electrochemical parameters such as electrolyte concentration, current density, voltage, and the resistance of the cell are calculated by the PP2D model according to the initial values. Secondly, the current distribution is calculated based on the voltage and resistance of the cells. Thirdly, the three-dimensional temperature distribution in the pack is obtained by using the FEM. Finally, the temperature distribution is fed back to the PP2D model to update the electrochemical parameters in the PP2D model.

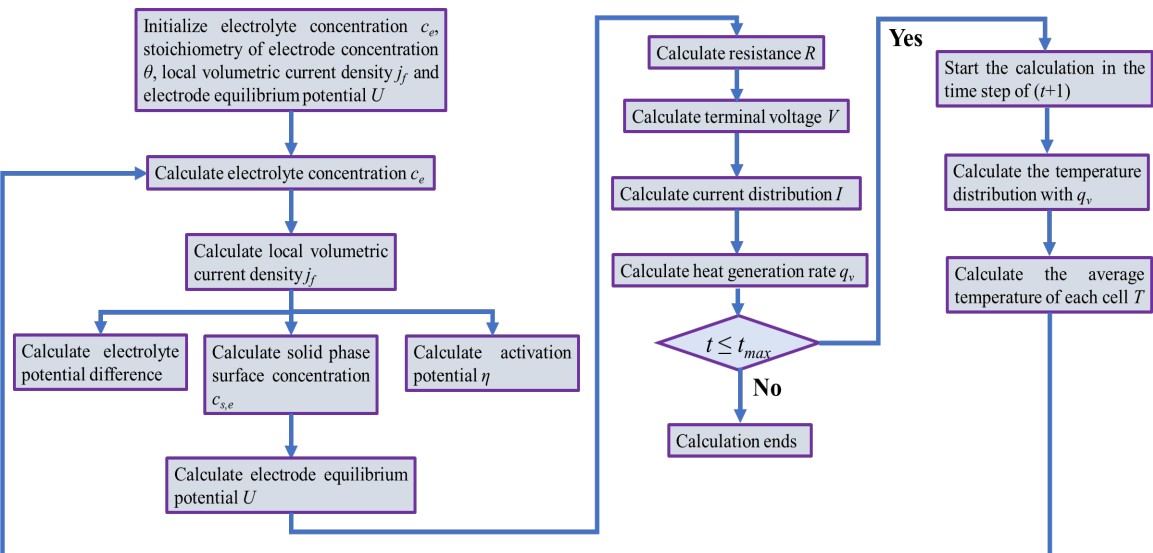

**Figure 6.** Solution process of electrochemical-thermal model.

## 3. Experimental Validation

### 3.1. Experimental Devices

Figure 7 presents the experimental devices for verifying the proposed model for the pack. The battery tester was used to charge/discharge the battery and measure its terminal voltage and current. Because the cell and module have different voltages, two battery testers were employed. The NEWARE BTS-8000 battery tester was for the cell, while the NEWARE CE7001 battery tester was for the pack. The error of the battery tester was ±0.1% for both current and voltage. A thermostat was used to control the ambient temperature for the cell and the pack. Pt100 platinum thermal resistance detector (PRTD) was used as the temperature sensor, and its signal was collected by using the Keysight 34970A temperature collector. The test error of PRTD is 0.15 °C. Figure 8 shows the arrangement of the temperature sensor. For the cell, three PRTDs were fixed at the center of the cell surface; for the pack, 24 PRTDs were placed on both sides of twelve cells. The pack used for the validation was composed of twelve 50 Ah NMC/graphite cells. Four PCBSs were connected in a series, and three cells were connected in parallel in each PCBS. The capacity of the pack was 150 Ah, and 1 C was 150 A. In the cell test, the cell temperature was calculated by averaging the measured temperature from the three sensors; in the pack test, the cell temperature was achieved by averaging the temperature from the two sensors located on the side wall of the cell.

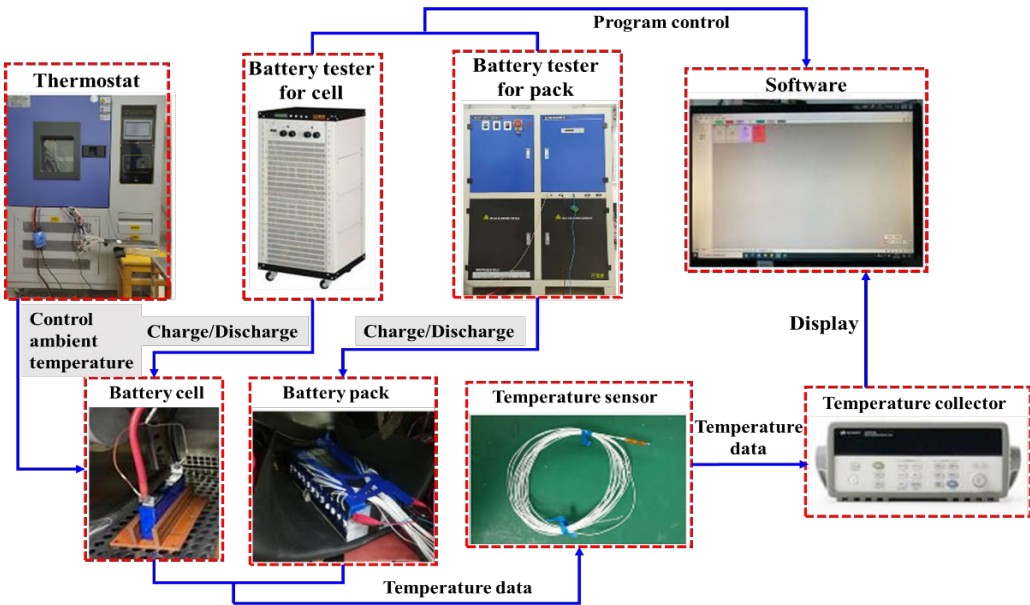

**Figure 7.** Experimental devices.

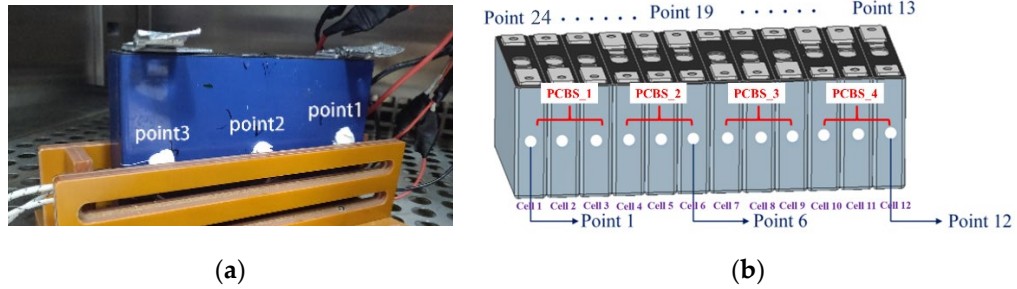

|   (**a**)   |   (**b**)   |

**Figure 8.** Location of temperature sensor: (**a**) in the cell test, (**b**) in the pack test.

### 3.2. Verification Results

The electrochemical-thermal model of cell was finite element method (FEM). The predicted terminal voltage and cell temperature were validated against the test data at the ambient temperatures of 10 °C, 20 °C, and 30 °C and against the discharging rates of 0.5 C, 1 C, and 2 C. According to the verification results given in Figure 9, the predicted terminal voltage and temperature develop almost the same as those measured by the experiment. The maximum difference of voltage is 0.08 V, and that of the temperature is 1.1 °C, indicating that the electrochemical-thermal model is accurate and can be used to describe the electrical and thermal performance of the cell.

The predicted terminal voltage of the pack and the temperatures of the cells were verified at different ambient temperatures with a 1C discharge rate. In order to avoid overdischarging, the discharge starts at SOC = 0.9 and ends at SOC = 0.1. According to the validation results of the pack presented in Figure 10, the simulated terminal voltage and the temperature have a similar evolution to the test value. The maximum difference of the voltage is 0.06 V, and that of the temperature is 1.2 °C. Those small errors prove that the proposed model can precisely predict the pack's electrical and thermal performance.

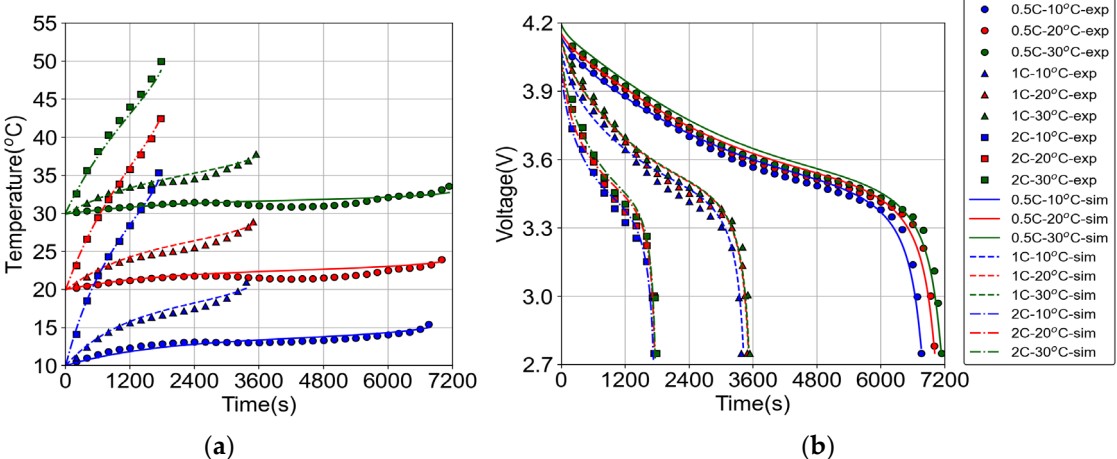

**Figure 9.** The verification of single battery under different discharge rate and different ambient temperature. (**a**) Temperature increase; (**b**) terminal voltage.

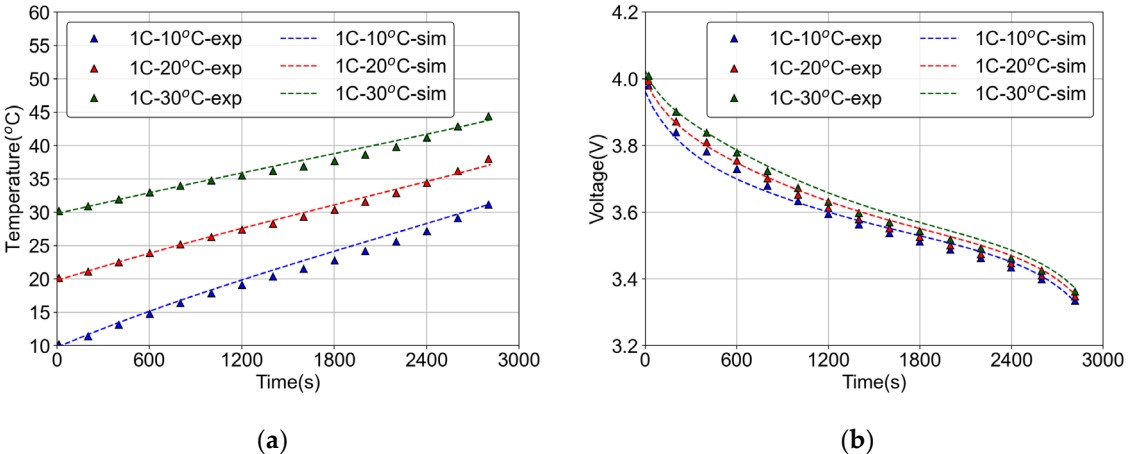

**Figure 10.** The verification of battery pack at different ambient temperature with 1C discharge rate: (**a**) temperature increase, (**b**) terminal voltage.

## 4. Influence of Inhomogeneous Cooling on Pack

### 4.1. Uneven Cooling Cases

Table 5 shows the cases of the inhomogeneous cooling. To produce the temperature distribution, the heat-transfer coefficient from 5 W/(K·m$^2$) to 220 W/(K·m$^2$) and coolant temperature ($T_c$) from 10 °C to 30 °C are imposed on both sides of the pack. The other surfaces of the pack are assumed to be adiabatic. When $h$ is ≤32 W/(K·m$^2$), air cooling is applied. When $h$ is ≥100 W/(K·m$^2$), water cooling is adopted. Moreover, in the simulation, the coolant temperature is considered the same as the pack initial temperature. The discharging rate is 1 C, and the SOC range is from 1 to 0.

**Table 5.** Parameters of inhomogeneous cooling.

| Case | $T_c$ (°C) | $h$ (W/K·m$^2$) | Case | $T_c$ (°C) | $h$ (W/K·m$^2$) |
|---|---|---|---|---|---|
| 1 | 10 | 5 | 9 | 30 | 100 |
| 2 | 20 | 5 | 10 | 10 | 175 |
| 3 | 30 | 5 | 11 | 20 | 175 |
| 4 | 10 | 32 | 12 | 30 | 175 |
| 5 | 20 | 32 | 13 | 10 | 220 |
| 6 | 30 | 32 | 14 | 20 | 220 |
| 7 | 10 | 100 | 15 | 30 | 220 |
| 8 | 20 | 100 | | | |

### 4.2. Temperature Distribution of the Pack

Figure 11 gives the temperature increase of cells in the pack due to the fact PCBS_1 and PCBS_4 are arranged on both sides of the pack, closer to the heat sink. Therefore, temperature variation in both PCBS_1 and PCBS_4 are wide, while that in PCBS_2 and PCBS_3 are narrow. As the battery arrangement and heat dissipation arrangement in the battery pack are symmetrically distributed, the average temperature and discharge current of the battery in the symmetrical position are the same. When the discharge is over, the temperature difference reaches the maximum. Maximum temperature differences in both PCBS_1 and PCBS_4 come to 10.64 °C, while that in PCBS_2 and PCBS_3 is only 1.84 °C. The large temperature difference caused a large difference in electrochemical parameters between the cells in these PCBSs and finally led to a large difference in current and SOC between the parallel-connected cells.

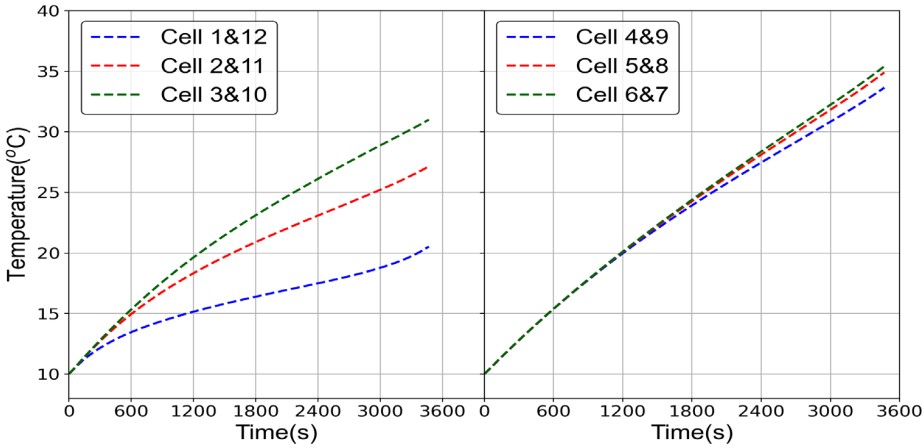

**Figure 11.** Temperature increase curve of the cells under the inhomogeneous cooling case 13.

Tables 6 and 7, respectively, show the maximum temperature difference ($\Delta T_{max\_diff}$) between cells and the average temperature of the package ($T_{ave\_pack}$) at the end of discharge. According to the information in the table, with the increasing of heat-transfer coefficient, temperature difference inside the pack increases and the temperature of the pack decreases. For example, when $T_c$ = 10 °C, $\Delta T_{max\_diff}$ increases to 10.64 °C, and $T_{ave\_pack}$ drops to 31.06 °C with the increase of $h$ because the temperature gradient of the pack is required to be wide to quickly transfer heat. Therefore, better cooling will cause lower $T_{ave\_pack}$, but the temperature between batteries changes greatly, which greatly affects the consistency of SOC between batteries.

**Table 6.** Maximum temperature difference between cells (°C).

|           | $T_c$ = 10 °C | $T_c$ = 20 °C | $T_c$ = 30 °C |
|-----------|-----------|-----------|-----------|
| $h = 5$   | 0.95      | 0.80      | 0.72      |
| $h = 32$  | 4.50      | 3.93      | 3.45      |
| $h = 100$ | 8.32      | 7.29      | 6.40      |
| $h = 175$ | 10.03     | 8.78      | 7.70      |
| $h = 220$ | 10.64     | 9.30      | 8.15      |

**Table 7.** Volume-weighted average temperature of pack after discharge (°C).

|           | $T_c$ = 10 °C | $T_c$ = 20 °C | $T_c$ = 30 °C |
|-----------|-----------|-----------|-----------|
| $h = 5$   | 36.66     | 42.75     | 49.36     |
| $h = 32$  | 34.72     | 40.91     | 47.89     |
| $h = 100$ | 32.49     | 38.96     | 46.19     |
| $h = 175$ | 31.44     | 38.05     | 45.39     |
| $h = 220$ | 31.06     | 37.72     | 45.11     |

### 4.3. Current Distribution of the Pack

Figure 12 presents the current distribution of the pack under the inhomogeneous cooling case 13. All four PCBSs in the battery pack have a similar pattern. The current of the cell in the middle of the PCBS approaches the average value, that is, one-third of total current passing the battery pack. The current of the other two batteries varies greatly, which is greater than or less than the average value. PCBS_1 and PCBS_4 have a large temperature difference, resulting in a large difference in their internal current. In the discharge process, the current of the battery with higher temperature in each PCBS first decreases from the average current in the early stage of discharge and then increases in the middle stage of discharge and gradually exceeds the average current, becoming the larger current in the battery pack. Therefore, the current difference in the battery pack first increases, then decreases, and then increases. As shown in Figure 12, the maximum current difference of the battery pack at 1 C is 16.87 A, which is 33.74% of the average value. Furthermore, the large current difference leads to the inconsistency of SOC between cells.

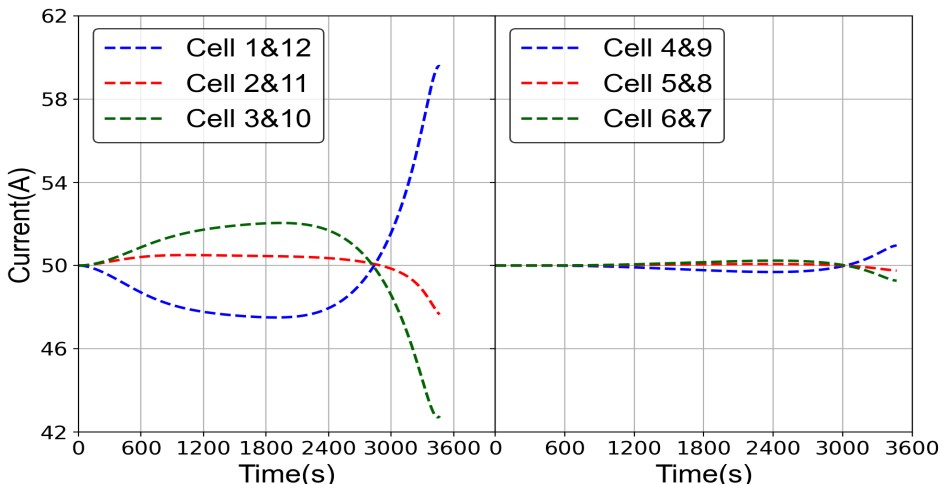

**Figure 12.** Current distribution of the pack under the inhomogeneous cooling case 13.

Table 8 shows the maximum current difference ($\Delta I_{max\_diff}$) between batteries in the battery pack. With the heat transfer progressively strengthening, current difference increases. When $T_c = 10\ ^\circ$C, $\Delta I_{max\_diff}$ grows by 1506.2% with the increase of $h$. The results show that the enhanced cooling of can lower the temperature, but it will cause poor temperature homogeneity and a large current gap in the parallel branch in the battery pack.

**Table 8.** Maximum current difference (A).

|  | $T_c = 10\ ^\circ$C | $T_c = 20\ ^\circ$C | $T_c = 30\ ^\circ$C |
| --- | --- | --- | --- |
| $h = 5$ | 1.12 | 0.75 | 0.51 |
| $h = 32$ | 5.99 | 4.04 | 2.70 |
| $h = 100$ | 12.45 | 8.38 | 5.57 |
| $h = 175$ | 15.70 | 10.57 | 7.01 |
| $h = 220$ | 16.87 | 11.36 | 7.53 |

### 4.4. SOC Distribution of the Pack

Figure 13 shows the SOC of cells under the inhomogeneous cooling case 13. Since the current in PCBSs determines the SOC of each battery, the SOC distribution in PCBSs has a likeness to the current distribution. Owning to the big current non-uniformity in PCBS_1 and PCBS_4, the batteries in these PCBSs have a bigger SOC gap than those in PCBS_2 and PCBS_3. As shown in Figure 14, the maximum SOC difference ($\Delta SOC_{max\_diff}$) in PCBS_1 is 4.89%, and that of PCBS_2 is only 0.40%. In each PCBS, the SOC decreases

with the expansion of the distance from the battery to the coolant, which is because of the bigger current caused by the higher temperature.

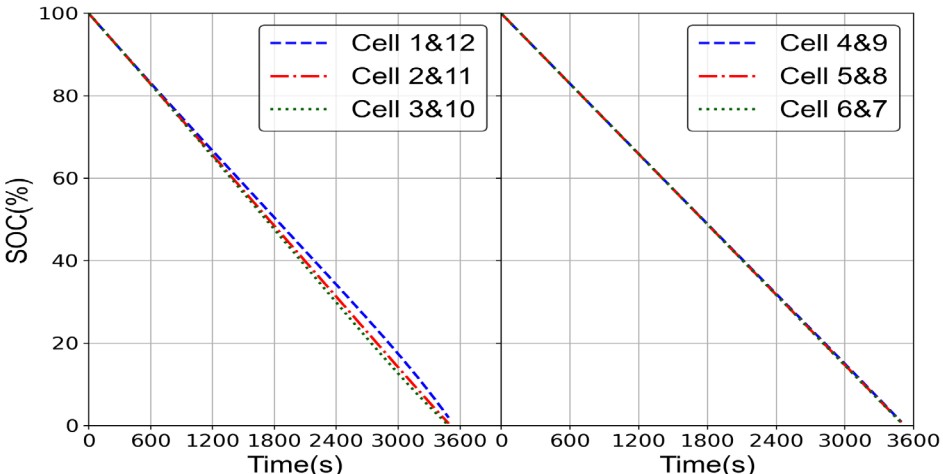

**Figure 13.** SOC of cells under the inhomogeneous cooling case 13.

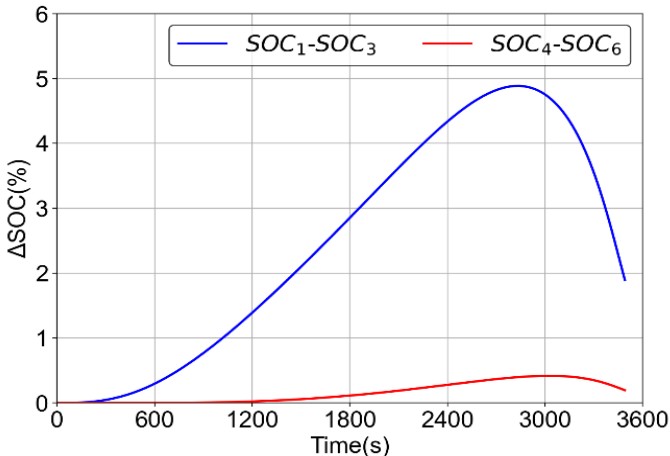

**Figure 14.** SOC difference of cells under the inhomogeneous cooling case 13.

Table 9 gives the maximum SOC difference ($\Delta SOC_{max\_diff}$). During the discharging process, it can be seen from Table 9 that due to the huge current change caused by cooling, the coolant with bigger $h$ has a larger SOC non-uniformity on the module. When $T_c = 10\ °C$, $\Delta SOC_{max\_diff}$ increases by 1530% with the increase of $h$.

**Table 9.** Maximum SOC difference (%).

|  | $T_c = 10\ °C$ | $T_c = 20\ °C$ | $T_c = 30\ °C$ |
| --- | --- | --- | --- |
| $h = 5$ | 0.30 | 0.21 | 0.14 |
| $h = 32$ | 1.65 | 1.10 | 0.73 |
| $h = 100$ | 3.52 | 2.31 | 1.52 |
| $h = 175$ | 4.52 | 2.95 | 1.92 |
| $h = 220$ | 4.89 | 3.19 | 2.07 |

Figure 15 depicts the relationship of coolant temperature $T_c$, the maximum temperature difference $\Delta T_{max\_diff,}$ and the maximum SOC difference $\Delta SOC_{max\_diff}$ under the inhomogeneous cooling case 13. For the pack designed as the maximum 1C discharge rate, the cell equalization goal of the control objective is to let $\Delta SOC_{max\_diff}$ less than 2%. Consequently, take $\Delta SOC_{max\_diff} = 2\%$ as the standard to determine the temperature difference of cells

in the pack, hoping objective $\Delta T_{max\_diff}$ can always make $\Delta SOC_{max\_diff}$ remain less than 2%. In Figure 16, as $T_c$ increases from 10 °C to 30 °C, the objective $\Delta T_{max\_diff}$ boundary increases from 5.21 °C to 7.94 °C. The result illustrates that the objective $\Delta T_{max\_diff}$ for the cell equalization is not a stable value but increases with the coolant temperature.

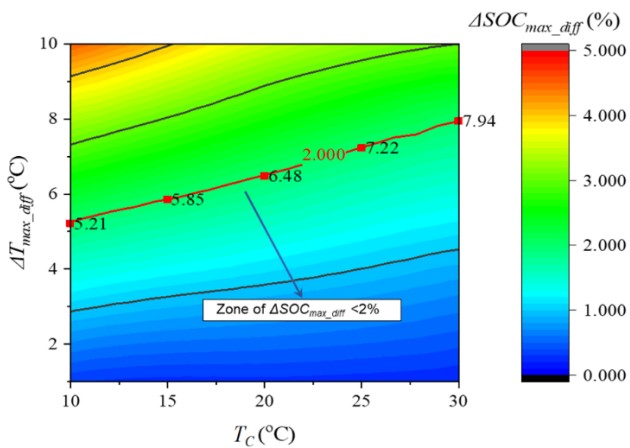

**Figure 15.** Relationship among $T_c$, $\Delta T_{max\_diff}$ and $\Delta SOC_{max\_diff}$ under the inhomogeneous cooling case 13.

### 4.5. Electrochemical Parameters of the Pack

Under the condition of uneven heat dissipation, the current of each battery in PCBSs is different, resulting in uneven battery SOC. Therefore, it is necessary to study how the electrochemical parameters affect the current distribution. Furthermore, according to Formula (61), in order to meet the equation conditions, the current passing through the battery with low terminal voltage $V$ will be smaller. Therefore, the study of influence of electrochemical parameters on current distribution can be converted to the study of influence of electrochemical parameters on terminal voltage changes.

From Formula (54), we can see the influencing factors of terminal voltage $V$. For convenience of interpretation, make the open-circuit voltage $U_{OCV} = U_p - U_n$, the electrolyte potential difference $\Delta\phi_e = \phi_{e,n}(0) - \phi_{e,p}(0)$, and the activation overpotential difference $\Delta\eta = \eta_n(0) - \eta_p(0)$. In this paper, the $R_{SEI}$ of formula (54) is 0, so the terminal voltage $V$ is only affected by the open-circuit voltage $U_{OCV}$, the electrolyte potential difference $\Delta\phi_e$, and the activation overpotential difference $\Delta\eta$. Formula (54) is simplified to $V = U_{OCV} - \Delta\phi_e - \Delta\eta$.

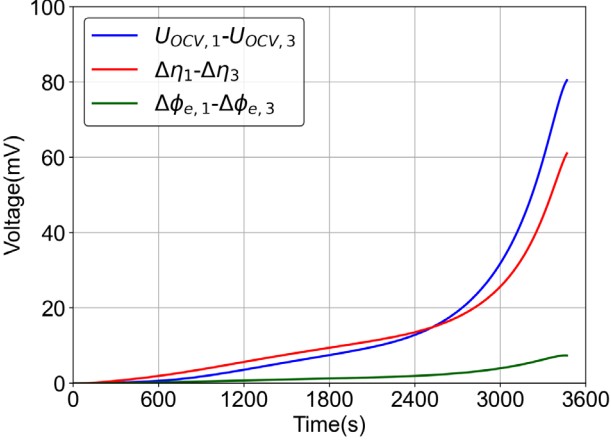

**Figure 16.** The difference of open-circuit voltage, the activation overpotential difference, and the electrolyte potential difference between cell 1 and cell 3 under the inhomogeneous cooling case 13.

In PCBSs, the cells with the largest temperature difference are $cell_1$ and $cell_3$. Figure 16 shows the difference of open-circuit voltage ($U_{OCV,1} - U_{OCV,3}$), the difference of the activation overpotential difference ($\Delta\eta_1 - \Delta\eta_3$), and the difference of the electrolyte potential difference ($\Delta\phi_{e,1} - \Delta\phi_{e,3}$) between cell 1 and cell 3 under the inhomogeneous cooling case 13.

As shown in Figure 16, compared with the difference of the open-circuit voltage ($U_{OCV,1} - U_{OCV,3}$) and the difference of the activation overpotential difference ($\Delta\eta_1 - \Delta\eta_3$), the difference of the electrolyte potential difference ($\Delta\phi_{e,1} - \Delta\phi_{e,3}$) is one order of magnitude smaller, so it can be ignored. Therefore, the change of terminal voltage is mainly determined by the open-circuit voltage and the activation overpotential difference.

It can be seen from the calculation formula in Table 4 that the $U_{OCV}$ is determined by SOC. The activation overpotential is calculated by Formula (26), so it is mainly affected by the electrochemical parameters $j_f$ and $i_0$. The activation overpotential $\eta$ is positively correlated with $j_f$ and negatively correlated with $i_0$. Similarly, due to the negative sign in front of $\eta_p(0)$, which is just offset by the negative signs of $j_{f,p}$, the activation overpotential difference $\Delta\eta$ is positively correlated with the absolute value of $j_f$ and negatively correlated with $i_0$.

Figure 17 shows the $j_f$ of PCBS_1 under the inhomogeneous cooling case 13. It can be seen from Formula (42) that $j_f$ is mainly affected by the current. This is also consistent with the results obtained. As shown in the Figure 16, the changing trend of the absolute value of $j_f$ is consistent with the current but is little affected by temperature.

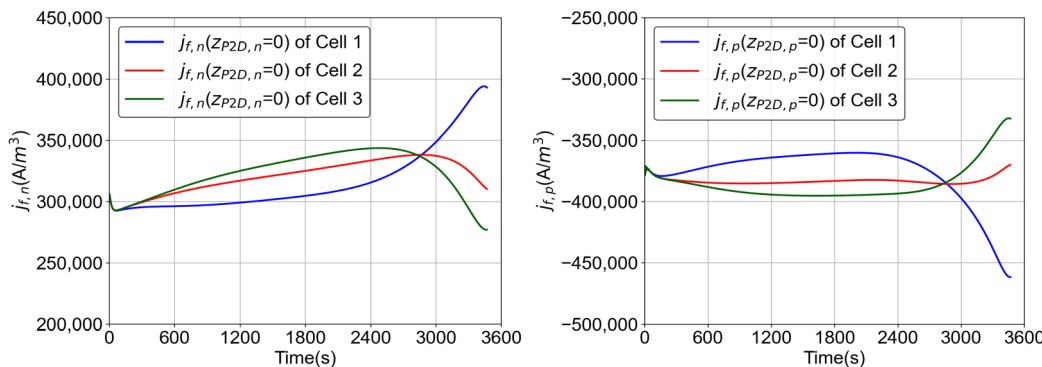

**Figure 17.** $j_f$ of PCBS_1 under the inhomogeneous cooling case 13.

Tables 10 and 11 display the maximum local volumetric current density difference ($\Delta j_{f,max\_diff}$) in the discharge process. As seen from Tables 10 and 11, because of the huge current change caused by cooling, the coolant with a large heat-transfer coefficient has large $\Delta j_{f,max\_diff}$. When $T_c = 10\,°C$, $\Delta j_{f,n,max\_diff}$ and $\Delta j_{f,p,max\_diff}$ increase by 1438% and 1414%, respectively, with the increase of $h$.

**Table 10.** Maximum local volumetric current density difference at the negative electrode ($\Delta j_{f,n,max\_diff}$) in the discharge process ($10^4$ A/m$^3$).

|  | $T_c = 10\,°C$ | $T_c = 20\,°C$ | $T_c = 30\,°C$ |
|---|---|---|---|
| $h = 5$ | 0.76 | 0.53 | 0.34 |
| $h = 32$ | 4.10 | 2.75 | 1.81 |
| $h = 100$ | 8.55 | 5.72 | 3.75 |
| $h = 175$ | 10.81 | 7.23 | 4.73 |
| $h = 220$ | 11.62 | 7.78 | 5.09 |

**Table 11.** Maximum local volumetric current density difference at the positive electrode ($\Delta j_{f,p,max\_diff}$) in the discharge process ($10^4$ A/m$^3$).

|  | $T_c$ = 10 °C | $T_c$ = 20 °C | $T_c$ = 30 °C |
|---|---|---|---|
| $h = 5$ | 0.85 | 0.57 | 0.39 |
| $h = 32$ | 4.57 | 3.09 | 2.05 |
| $h = 100$ | 9.50 | 6.41 | 4.24 |
| $h = 175$ | 11.98 | 8.09 | 5.34 |
| $h = 220$ | 12.87 | 8.70 | 5.74 |

Figure 18 shows the $i_0$ of PCBS_1 under the inhomogeneous cooling case 13. It can be seen that $i_0$ increases with temperature increase, which is primarily due to the effect that reaction rate constant $k$ increases with temperature.

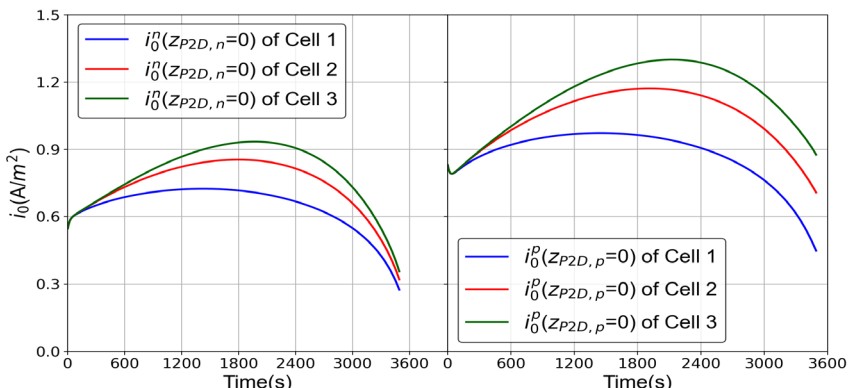

**Figure 18.** $i_0$ of PCBS_1 under the inhomogeneous cooling case 13.

Tables 12 and 13 give the maximum exchange current density difference ($\Delta i_{0,max\_diff}$) in the discharge process. It can be seen from Tables 12 and 13 that because of the huge current change caused by cooling, the coolant with a large heat-transfer coefficient has large $\Delta i_{0,max\_diff}$ on the module. When $T_c$ = 10 °C, $\Delta i_{0,n,max\_diff}$ and $\Delta i_{0,p,max\_diff}$ increase by 1100% and 875%, respectively, with the increase of $h$.

**Table 12.** Maximum exchange current density difference at the negative electrode ($\Delta i_{0,n,max\_diff}$) in the discharge process (A/m$^2$).

|  | $T_c$ = 10 °C | $T_c$ = 20 °C | $T_c$ = 30 °C |
|---|---|---|---|
| $h = 5$ | 0.02 | 0.02 | 0.03 |
| $h = 32$ | 0.11 | 0.11 | 0.12 |
| $h = 100$ | 0.19 | 0.20 | 0.23 |
| $h = 175$ | 0.22 | 0.24 | 0.27 |
| $h = 220$ | 0.24 | 0.26 | 0.29 |

**Table 13.** Maximum exchange current density difference at the negative electrode ($\Delta i_{0,p,max\_diff}$) in the discharge process (A/m$^2$).

|  | $T_c$ = 10 °C | $T_c$ = 20 °C | $T_c$ = 30 °C |
|---|---|---|---|
| $h = 5$ | 0.04 | 0.04 | 0.05 |
| $h = 32$ | 0.18 | 0.20 | 0.22 |
| $h = 100$ | 0.32 | 0.35 | 0.39 |
| $h = 175$ | 0.37 | 0.41 | 0.46 |
| $h = 220$ | 0.39 | 0.43 | 0.48 |

To sum up, the current distribution in the early stage of battery discharge is determined by temperature. The $i_0$ of the battery with lower temperature is lower, resulting in the

activation overpotential difference $\Delta \eta$ higher, which affects the gradual decrease of its discharge current. In the middle discharge stage of discharging, open-circuit voltage $U_{OCV}$ begins to play a leading role due to the gradually increasing SOC gap. Due to the higher SOC, the discharge current of the battery with lower temperature began to increase and gradually became a larger current in the later stage of discharge, and the SOC gap between batteries began to narrow. In the later stage of discharging, the battery with lower temperature has higher current, so its $i_0$ is lower and its $j_f$ is higher, resulting in a larger activation overpotential difference $\Delta \eta$. Although the SOC gap between batteries is narrowing, the $U_{OCV}$ accelerates to decrease with the decrease of SOC at the later stage of discharging. As a result, the open-circuit voltage gap between batteries gradually becomes larger, and the $U_{OCV}$ still plays a leading role. The SOC of batteries with lower temperature is higher, and the discharge current continues to increase due to higher $U_{OCV}$. Therefore, in general, the discharge current of the battery with lower temperature first decreases in the early stage of discharge, increases in the middle stage of discharge, gradually exceeds the current of the battery with higher temperature, and becomes a larger current in the late stage of discharge.

As for SOC, the difference between the battery with lower temperature and the battery with higher temperature first increases and then decreases. While cooling the battery pack with coolant can reduce the temperature, this will lead to poor temperature homogeneity of the parallel branch. The increasing of the temperature difference will lead to a greater difference in the electrochemical parameters, which will affect the voltage and current and ultimately affect the greater unevenness of the battery SOC.

## 5. Conclusions

In this paper, the electrochemical thermal coupling model of the parallel battery pack was established. Then, it is used to study the uneven temperature distribution and the coupling relation between electrical and electrochemical parameters in the battery pack under different heat dissipation conditions. The mechanism of how the battery pack temperature difference affects the SOC distribution was also clearly found. According to the results, the following conclusions can be drawn.

(1) The temperature difference in the battery pack caused by cooling led to the inconsistency of SOC. It is found that the discharge current of the cell with lower temperature decreases at the initial stage of discharge, increases at the middle stage of discharge, and gradually exceeds the current of the battery with higher temperature to become a larger current in the late stage of discharge. The current distribution at the early stage of battery discharge is determined by temperature. The $i_0$ of the battery with lower temperature is lower, resulting in the activation overpotential difference $\Delta \eta$ higher, which affects the gradual reduction of its discharge current. The open-circuit voltage $U_{OCV}$ plays a leading role in the middle and late stages of discharge. The open-circuit voltage $U_{OCV}$ of batteries with higher SOC is higher, so the discharge current of batteries with lower temperature is further increased. During the whole discharge process, the SOC gap in the battery pack first increases and then decreases.

(2) Good cooling can reduce the average temperature of the battery pack, but it will cause a huge temperature gradient inside the pack. In addition, an overly high temperature gradient inside the battery pack will affect the electrochemical parameters, thus making the current uniformity worse. When $T_c = 10\ °C$, $\Delta T_{max\_diff}$ increases by 1120% and $\Delta I_{max\_diff}$ increases by 1506.2% with the increase of $h$.

(3) Good cooling brings great SOC non-uniformity to the battery pack. When $T_c = 10\ °C$, $\Delta SOC_{max\_diff}$ increases by 1530% with the increase of $h$. Therefore, the control of battery pack temperature and temperature difference is contradictory and should be balanced. In addition, when the initial temperature, coolant temperature, and ambient temperature are set to the same value, the objective maximum temperature difference $\Delta T_{max\_diff}$ most suitable for controlling the uniformity of SOC increases with the increase of $T_c$. When $T_c$

increases from 10 °C to 30 °C, the objective $\Delta T_{max\_diff}$ boundary to maintain $\Delta SOC_{max\_diff}$ within 2% increases from 5.21 °C to 7.94 °C.

**Author Contributions:** Conceptualization, Z.D. and Y.X.; methodology, Z.D. and X.M.; software, X.M.; validation, Y.X., X.M. and Z.D.; investigation, K.Z., B.C. and Y.F.; writing—original draft preparation, X.M.; writing—review and editing, Y.X.; visualization, Y.X. and X.M.; supervision, K.Z., B.C. and Y.F. All authors have read and agreed to the published version of the manuscript.

**Funding:** This research was funded by the National Natural Science Foundation of China (Grant No. 52102420). The authors gratefully acknowledge the great help of the fund.

**Data Availability Statement:** Not applicable.

**Conflicts of Interest:** The authors declare no conflict of interest.

## Nomenclature

| | |
|---|---|
| $A$ | Electrode plate area, m$^2$ |
| $\alpha_s$ | Specific interfacial surface area, m$^{-1}$ |
| $c_e$ | Electrolyte concentration, mol m$^{-3}$ |
| $c_{e,0}$ | Average electrolyte concentration, mol m$^{-3}$ |
| $c_s$ | Solid-phase concentration, mol m$^{-3}$ |
| $c_{s,e}$ | Solid-phase surface concentration, mol m$^{-3}$ |
| $\bar{c}$ | Volume-averaged solid-phase concentration, mol m$^{-3}$ |
| $D_s$ | Solid-phase diffusivity, m$^2$s$^{-1}$ |
| $D_e$ | Electrolyte phase diffusivity, m$^2$s$^{-1}$ |
| $F$ | Faraday's constant, 96,487 C mol$^{-1}$ |
| $I$ | Current, A |
| $i_0$ | Exchange current density, A m$^{-2}$ |
| $j$ | Reaction flux, mol m$^{-2}$ s$^{-1}$ |
| $j_f$ | Local volumetric current density, A m$^{-3}$ |
| $k$ | Reaction rate, mol$^{-0.5}$m$^{2.5}$s$^{-1}$ |
| $L$ | Thickness, m |
| $Q$ | Total amount of lithium-ion in each region, mol m$^{-2}$ |
| $\bar{q}$ | Volume-averaged concentration flux, mol m$^{-4}$ |
| $R$ | Battery resistance, $\Omega$ |
| $R_s$ | Particle radius, m |
| $T$ | Temperature, K |
| $r$ | Welding resistance, $\Omega$ |
| $t$ | Time, s |
| $t^+$ | Lithium-ion transfer number |
| $U$ | Electrode equilibrium potential, V |
| $U_{OCV}$ | Open-circuit voltage, V |
| $V$ | Terminal voltage of battery, V |
| $x$ | lD coordinate across the cell, m |
| $z$ | lD coordinate across electrode/separator, m |
| $\varepsilon_s$ | Active material volume fraction |
| $\varepsilon_e$ | Electrolyte volume fraction |
| $\theta$ | Stoichiometry of electrode concentration |
| $\theta_0$ | Stoichiometry at 0% SOC |
| $\theta_{100}$ | Stoichiometry at 100% SOC |
| $\alpha_a, \alpha_c$ | Charge transfer coefficient |
| $\sigma$ | Solid-phase conductivity, S m$^{-1}$ |
| $\kappa$ | Electrolyte phase conductivity, S m$^{-1}$ |
| $\varphi_s$ | Solid-phase potential, V |
| $\varphi_e$ | Electrolyte phase potential, V |
| $\eta$ | Activation overpotential, V |
| $dlnf_{\pm}/dlnc_e$ | Activity dependence |
| $\Delta t$ | Sampling step, s |

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
