# Peer review of "A Comprehensive Study on the Effect of Inhomogeneous Heat Dissipation on Battery Electrochemical Performance"

_electronics, doi:10.3390/electronics12061266_

Round 1
Reviewer 1 Report
In this paper, the author proposed a Polynomial Approximation Pseudo-Two-Dimensional (PP2D) method to establish a battery pack model to simulate the temperature, current and SOC distribution over battery pack. The simulated results are validated by experimental data. According to the author's data, the temperature distribution was uneven due to the coolant and initial ambient temperature, which will cause the uneven current distribution. Furthermore, the uneven current led to SOC varying, which will further impact the electro-chemical performance.
The paper is very educated and interesting besides several minor typos:
(1) Line 442: "at different ambient temperature with different discharge rates of 1C". You may be meant to say"at different ambient temperature with different discharge rates" or "at different ambient temperature with 1C discharge rate"
(2) Fig. 10 Caption: Fig10 only showed the data at 1C discharge not different discharge rate
(3) Line 537 and line 542, it should be Fig. 14 instead of Fig.14
Beside the typos, I have one more question, this question is just for my knowledge: Line 561, you mentioned in this paper Rsei =0 , I am wondering why. The uneven temperature distribution generally will cause uneven SEI growth, which will contribute to the Rsei.
Author Response
In this paper, the author proposed a Polynomial Approximation Pseudo-Two-Dimensional (PP2D) method to establish a battery pack model to simulate the temperature, current and SOC distribution over battery pack. The simulated results are validated by experimental data. According to the author's data, the temperature distribution was uneven due to the coolant and initial ambient temperature, which will cause the uneven current distribution. Furthermore, the uneven current led to SOC varying, which will further impact the electro-chemical performance.
Reply: Thanks so much for your impressive comment and evaluation.
The paper is very educated and interesting besides several minor typos:
- Line 442: "at different ambient temperature with different discharge rates of 1C". You may be meant to say"at different ambient temperature with different discharge rates" or "at different ambient temperature with 1C discharge rate"
Reply: Thank you for your careful and detailed review. There are omissions in the expression of Line 442 in the original manuscript. The corresponding content of the article has been modified to "at different ambient temperature with 1C discharge rate" in Line 440 of the new manuscript.
Fig. 10 Caption: Fig10 only showed the data at 1C discharge not different discharge rate
Reply: Thanks very much for your meticulous review. There is a careless mistake in the title description of title in Figure 10. Figure 10 is intended to verify the simulation results of 1C discharge. The caption of Figure 10 has been rewritten as "The verification of battery pack at different ambient temperature with 1C discharge rate: a) temperature rise, (b) terminal voltage".Thank you again for your careful review.
- Line 537 and line 542, it should be Fig. 14 instead of Fig.14
Reply: Thank you for your detailed questions. The same number appears when numbering pictures in the original manuscript. The corresponding picture number in the new manuscript has been correctly modified.
- Beside the typos, I have one more question, this question is just for my knowledge: Line 561, you mentioned in this paper Rsei =0, I am wondering why. The uneven temperature distribution generally will cause uneven SEI growth, which will contribute to the Rsei.
Reply: It is a reasonable question. The reason for the Rsei =0 is that the Rsei value given in the reference [30] cited in Table 3 is equal to 0. The influence of Rsei is ignored in reference [30]. It’s true that the uneven temperature distribution generally will cause uneven SEI growth, which will contribute to the Rsei. The effect of non-uniform temperature on Rsei will be taken into account in future studies. Thank you again for your question.

Reviewer 2 Report
· Title of the investigation is adequate.
· The noteworthy results of this investigation are adequate.
· The plotted curves for the physical parameters are adequate.
· The significance of the study is adequate.
· Impact of the results looking correct.
· Over all this investigation is well framed and well written.
· Discussion part has enough physical interpretations.
Author Response
Reviewer #2
Title of the investigation is adequate.
The noteworthy results of this investigation are adequate.
The plotted curves for the physical parameters are adequate.
The significance of the study is adequate.
Impact of the results looking correct.
Over all this investigation is well framed and well written.
Discussion part has enough physical interpretations.
Reply: Thanks so much for your impressive comment and evaluation.

Reviewer 3 Report
In this paper, the unbalanced discharge of lithium-ion battery module caused by heat dissipation is studied both theoretically and experimentally.
This is a systematic and detailed study, worth publication after some minor revisions.
1. Sub-figure definition can make Figures clearer. For example, two sub-figures of Figure 9 can be defined as a and b, respectively.
2. Experimental and simulation results of Figure 3 and others can be presented with the same color for better understanding.
3. For Figures with x-axis as time, the x-axes should begin from 0 as physically negative time is meaningless.
4. Some more parameters can be symbolized, such as open circuit voltage for simplified expression.
Author Response
Reviewer #3
In this paper, the unbalanced discharge of lithium-ion battery module caused by heat dissipation is studied both theoretically and experimentally.
This is a systematic and detailed study, worth publication after some minor revisions.
Reply: Thanks so much for your impressive comment and evaluation.
Sub-figure definition can make Figures clearer. For example, two sub-figures of Figure 9 can be defined as a and b, respectively.
Reply: It is a reasonable review. The two sub-figures of Figure 9 in the manuscript are defined as a and b respectively. The sub-figure definition can make Figures clearer. Similarly, we also defined the two sub-figures of Figure 10 as a and b.
Experimental and simulation results of Figure 3 and others can be presented with the same color for better understanding.
Reply: Thank you for your constructive suggestions. The figures in the manuscript have been modified, and the experimental and simulation curves in the verification figures are represented by the same color. The modified figures are more intuitive.
- For Figures with x-axis as time, the x-axes should begin from 0 as physically negative time is meaningless.
Reply: Thank you for your detailed questions. The figures with the x-axis as the time in the manuscript have been modified. The x-axis of the modified figures starts from 0. We hope that the modified figures will meet your requirements.
- Some more parameters can be symbolized, such as open circuit voltagefor simplified expression.
Reply: Thanks very much for your meticulous review. The open circuit voltage in the manuscript is symbolized as UOCV, and the relevant content of the manuscript is simplified.

Reviewer 4 Report
This study provides useful insights into heat dissipation for battery systems.
Comments :
1. Suggesting authors revise the title as “ A comprehensive study on the effect of inhomogeneous heat dissipation on battery electrochemical performance”.
2. Suggesting authors check the manuscript thoroughly for spacing and grammatical errors.
3. Quality of the figures can be improved (including figure details size and color).

Author Response
Reviewer #4
This study provides useful insights into heat dissipation for battery systems.
Reply: Thanks so much for your impressive comment and evaluation.
- Suggesting authors revise the title as “A comprehensive study on the effect of inhomogeneous heat dissipation on battery electrochemical performance”.
Reply: Thanks for your advice. The title of the article has been revised to “A comprehensive study on the effect of inhomogeneous heat dissipation on battery electrochemical performance”.
- Suggesting authors check the manuscript thoroughly for spacing and grammatical errors.
Reply: It is a reasonable review. We carefully reviewed the spacing errors in the manuscript and invited a native English speaker to check the grammar errors. We hope it meets the requirements of Electronics now.
- Quality of the figures can be improved (including figure details sizeand color).
Reply: Thank you for your detailed questions. The figures in the manuscript have been modified, and the experimental and simulation curves in the verification figures are represented by the same color. In addition, the font size of the figures is increased to make the figures more intuitive and clear.
